EMBO
*reports*

# Interleukin-12/23 deficiency differentially affects pathology in male and female Alzheimer's disease-like mice

Pascale Eede[1],[††] (iD), Juliane Obst[1],[†],[††], Eileen Benke[1], Genevieve Yvon-Durocher[1], Bernhard C Richard[1], Niclas Gimber[2] (iD), Jan Schmoranzer[2] (iD), Annett Böddrich[3], Erich E Wanker[3] (iD), Stefan Prokop[1],[‡],[§],[¶],[††] (iD) & Frank L Heppner[1],[4],[5],[6],[*],[††] (iD)

## Abstract

Pathological aggregation of amyloid-β (Aβ) is a main hallmark of Alzheimer's disease (AD). Recent genetic association studies have linked innate immune system actions to AD development, and current evidence suggests profound gender differences in AD pathogenesis. Here, we characterise gender-specific pathologies in the APP23 AD-like mouse model and find that female mice show stronger amyloidosis and astrogliosis compared with male mice. We tested the gender-specific effect of lack of IL12p40, the shared subunit of interleukin (IL)-12 and IL-23, that we previously reported to ameliorate pathology in APPPS1 mice. IL12p40 deficiency gender specifically reduces Aβ plaque burden in male APP23 mice, while in female mice, a significant reduction in soluble $Aβ_{1-40}$ without changes in Aβ plaque burden is seen. Similarly, plasma and brain cytokine levels are altered differently in female versus male APP23 mice lacking IL12p40, while glial properties are unchanged. These data corroborate the therapeutic potential of targeting IL-12/IL-23 signalling in AD, but also highlight the importance of gender considerations when studying the role of the immune system and AD.

**Keywords** Alzheimer's disease; gender; IL-12/IL-23; innate immunity; β-amyloid

**Subject Categories** Immunology; Molecular Biology of Disease; Neuroscience

## Introduction

Alzheimer's disease (AD) is a chronic progressive neurodegenerative disorder associated with extracellular and intracellular protein aggregates [1] which induce synaptic dysfunction and degeneration of neurons and cause a characteristic clinical syndrome with prominent cognitive impairment [2,3]. Extracellular amyloid-β (Aβ) deposits are one of the prominent hallmarks of the disease [1], and a dysregulation of Aβ metabolism is thought to be one of the earliest pathological changes observable in AD patients, decades before first clinical symptoms occur [4,5]. Mouse models of Aβ deposition provide a useful tool to study amyloidogenesis *in vivo* [6] which in these mouse models is accompanied by an activation of the innate immune response characterised by activated microglia surrounding Aβ plaques [7], mimicking microglia activation observed in the brains of AD patients [8].

In the last decade, the importance of the innate immune response in AD pathogenesis has risen, driven by the discovery of multiple variants in immune system-associated genes conferring an increased risk for the development of sporadic AD, including the microglia cell surface receptors TREM2 and CD33 [9,10]. Even though these data suggest that the innate immune system plays an important role in AD, the exact nature of this immune response and its impact on disease is still far from clear [7,11]. While the short-term depletion of microglia had no major impact on development or progression of Aβ pathology [12,13], suggesting that microglia are rather inefficient in acutely regulating Aβ load, there are numerous examples in which modulation of the microglia response towards Aβ did have a

1   Department of Neuropathology, corporate member of Freie Universität Berlin, Humboldt-Universität zu Berlin, Berlin Institute of Health, Charité – Universitätsmedizin Berlin, Berlin, Germany
2   Advanced Medical Bioimaging Core Facility (AMBIO), corporate member of Freie Universität Berlin, Humboldt-Universität zu Berlin, Berlin Institute of Health, Charité - Universitätsmedizin Berlin, Berlin, Germany
3   Neuroproteomics, Max Delbrück Center for Molecular Medicine, Berlin, Germany
4   Cluster of Excellence, NeuroCure, Berlin, Germany
5   Berlin Institute of Health (BIH), Berlin, Germany
6   German Center for Neurodegenerative Diseases (DZNE) Berlin, Berlin, Germany
    *Corresponding author. Tel: +49 30 450 536 032; E-mail: frank.heppner@charite.de
    ††These authors contributed equally to this work
    †Present address: ARUK Oxford Drug Discovery Institute, University of Oxford, Oxford, UK
    ‡Present address: Department of Pathology, University of Florida, Gainesville, FL, USA
    §Present address: Center for Translational Research in Neurodegenerative Disease, University of Florida, Gainesville, FL, USA
    ¶Present address: Fixel Institute for Neurological Diseases, University of Florida, Gainesville, FL, USA

major impact on disease progression [14–16], especially in long-term depletion settings [17]. Similarly, we have reported an upregulation of the inflammatory cytokines interleukin (IL)-12 and IL-23 by microglia in the brain of AD-like APPPS1 mice and demonstrated that targeting IL12p40, the common subunit of both IL-12 and IL-23, using either genetic or pharmacological strategies reduced Aβ pathology and ameliorated cognitive deficits inherent in these mice [18]. The finding that IL12p40 levels are also de-regulated in the cerebrospinal fluid (CSF) of AD patients [18], the correlation between IL12p40 levels in plasma and mild cognitive impairment (MCI) and AD [19] and the detection of elevated levels of IL12p70 in brain tissue of AD patients [20], further emphasises the relevance of these pathways for the human disease condition.

Another observation derived from epidemiological studies implies differences in the prevalence to develop AD between male and female subjects [21–23]. Additionally, there are known differences in both the innate and adaptive immune responses between males and females [24], including gender-specific differences in male versus female microglial phenotypes [25–27]. Single-cell transcriptome analyses confirmed these notions by finding a gender-specific response within all brain cell populations of male and female AD patients, including microglia [28]. In light of these observations, our study aimed at identifying whether the effect of targeting the microglia-expressed IL12p40 on disease pathogenesis is model- and/or gender-specific. We therefore crossed mice deficient in IL12p40 to yet another AD-like mouse model, namely APP23 mice [29]. APP23 mice show a much slower rate of Aβ deposition than APPPS1 mice [30] utilised in earlier studies, which recapitulates more closely Aβ pathology of human AD patients with respect to the Aβ accumulation time course and the histopathological Aβ composition consisting of a sound mixture of "soft"/"diffuse" and "core" Aβ plaques. Similar to effects described in human AD patient populations [21–23], gender differences in plaque deposition have been described in this mouse model, although these have not been characterised thoroughly to the best of our knowledge [31,32]. To address the latter, we assessed gender-specific properties of Aβ deposition as well as Aβ processing, surrogate markers of neuritic dystrophy and glial activation in male and female APP23 mice lacking or harbouring the IL-12/IL-23 signalling pathway.

## Results

### Female APP23 mice show increased Aβ pathology and astrogliosis compared to male mice

We and others have previously described an increase in the IL12p40 subunit shared by the cytokines IL-12 and IL-23, in AD-like mouse models [18,33], as well as CSF [18], plasma [19] and brain tissue [20] of AD patients, but so far, no analysis of potential gender differences has been performed. In order to validate these findings in a mouse model of AD with slow Aβ accumulation, more closely representing Aβ pathology and Aβ composition in sporadic AD patients, we made use of the APP23 mouse model harbouring the Swedish (KM670/671NL) mutation in the gene encoding the amyloid precursor protein (APP) [29]. The APP23 mouse model reportedly shows gender differences when examining Aβ plaque load and behavioural characteristics [31,32], which appears to relate more closely to the

pathogenetic events mimicking sporadic AD [21–23]. However, to our knowledge no gender-specific side-by-side comparison of the AD-like changes in male and female APP23 mice has been reported so far. We therefore aimed at quantifying late-stage plaque burden in 21-month-old APP23 male versus female mice using biochemical and histological methods.

In order to gain insights into Aβ accumulation, we generated consecutive protein homogenates with increased detergent stringency [34] of brains from male and female APP23 mice which were each analysed on the Meso Scale Diagnostics (MSD) platform to measure $A\beta_{1–40}$ and $A\beta_{1–42}$ content. Compared to male mice, we found that female APP23 mice contain twofold higher levels of the soluble (TBS fraction) and insoluble (SDS fraction) $A\beta_{1–40}$ isoform (Fig 1A) as well as the $A\beta_{1–42}$ isoform (Appendix Fig S1A). We could also confirm the published observation that $A\beta_{1–40}$ species are the main Aβ isoform present in brains of APP23 mice, while the more aggregation-prone $A\beta_{1–42}$ isoform is less abundant (Fig 1A; Appendix Fig S1A). Histologically, Aβ plaque burden can be characterised as being "diffuse" or "compact" [35]. Both types of plaques can be detected with antibodies targeting the Aβ protein itself, such as 4G8 or the chemical compound pFTAA [36]. Additionally, compact plaques can be visualised specifically using β-sheet-binding dyes such as Congo Red [37]. In APP23 mice, we observed that the cortical area covered by 4G8-, pFTAA- and Congo Red-positive amyloid-β plaques was twice as high in female mice compared to male mice (Fig 1B–D). Using the ratio between the area covered by 4G8-positive plaques and Congo Red-positive compact plaques, we determined that compact plaques only account for a quarter of total plaques in both male and female APP23 mice (Appendix Fig S1B).

To further investigate the differences between males and females in the APP23 model, a filter retardation assay was used to determine the presence of Aβ aggregates in protein lysates [38]. Protein homogenates from APP23 mouse brains were positively selected for aggregates larger than 0.2 μm in size. Immunostaining for 6E10, staining the Aβ protein, revealed that male mice have Aβ aggregates in the TBS-soluble protein fractions, few in the Triton-X fraction and none in the SDS fraction. However, an increased amount of aggregates was found in the FA fraction. In female mice, on the other hand, aggregates were found increasingly in the Triton-X fraction, SDS-soluble fraction and FA fraction at higher levels than in male mice (Fig 1E; Appendix Fig S1C). This indicates gender-specific Aβ aggregation properties and higher levels of insoluble Aβ aggregates in female mice. Neuritic dystrophy is another common pathological characteristic found in APP23 mice [29,39]. We thus stained tissue sections with BACE1, which has been suggested to act as a surrogate marker of neuritic dystrophy [40–43]. We noted that plaque-associated BACE1 immunoreactivity normalised to 4G8-positive Aβ plaques did not show any gender-specific differences (Fig 2A). The APP23 mouse strain also shows prominent microgliosis and astrocytosis around the amyloid deposits found in the brain [29,44–46]. Stereological analysis revealed a rise in the number of cortical astrocytes in female APP23 mice (Fig 2B), which correlated with increased Congo Red- and 4G8-positive plaque burden (Fig EV1A). To phenotype microglial characteristics, we quantified the number of plaque-associated microglia within 30 μm of the plaque border as well as their expression of Clec7a, which has been described as a marker of activated microglia in various disease contexts [47,48]. Both the number of plaque-associated microglia and the Clec7a

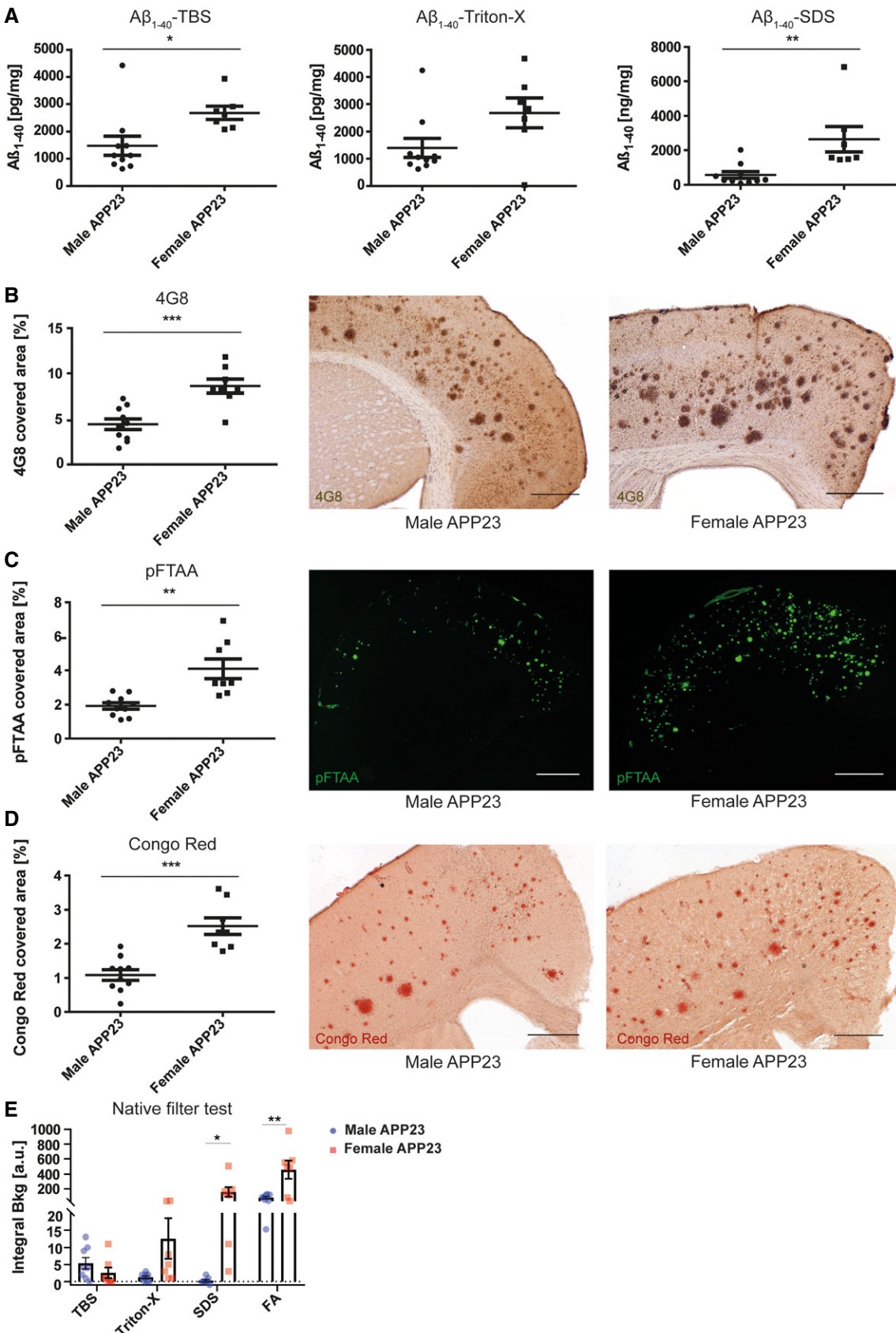

**Figure 1.**

**Figure 1.  Female APP23 mice at 21 months have higher Aβ burden than male APP23 mice.**

A   Quantitative analysis of the Aβ$_{1-40}$ protein in the TBS (*$P$ = 0.0212), Triton-X ($P$ = 0.0544) and SDS (**$P$ = 0.0063) fractions of brain homogenates from male ($n$ = 7) and female ($n$ = 10) APP23 mice. Mean ± SEM, statistical analysis: two-tailed unpaired $t$-test.

B   Stereological analysis of cortical area covered by 4G8-positive plaques in male ($n$ = 10) and female ($n$ = 8) APP23 mice (left) and representative images (right), scale bar = 500 μm. Mean ± SEM, statistical analysis: two-tailed unpaired $t$-test, ***$P$ = 0.0004.

C   Fluorescence intensity-based analysis of pFTAA-stained Aβ plaques in the cortex of male ($n$ = 10) and female ($n$ = 8) APP23 mice (left) and representative images (right), scale bar = 1 mm. Mean ± SEM, statistical analysis: two-tailed unpaired $t$-test, **$P$ = 0.0011.

D   Stereological analysis of cortical area covered by Congo Red-positive plaques in male ($n$ = 10) and female ($n$ = 8) APP23 mice (left) and representative images (right), scale bar = 500 μm. Mean ± SEM, statistical analysis: two-tailed unpaired $t$-test, ***$P$ = 0.0001.

E   Native filter assay analysis of TBS ($P$ = 0.2453), Triton-X ($P$ = 0.0604), SDS (*$P$ = 0.0196) and formic acid (FA) (**$P$ = 0.0057) fractions from male ($n$ = 8) and female ($n$ = 7) APP23 mouse brain homogenates. Mean ± SEM, statistical analysis: two-tailed unpaired $t$-test between the same fractions.

staining intensity within these cells were similar in male and female APP23 mice (Fig 2C). We used radial intensity profiling of 4G8-positive signal within Iba1-positive microglia as an indicator of microglial Aβ uptake. This analysis showed 4G8 intensity peaks inside the cell (~4 μm), but revealed no significant difference between both 4G8 traces, i.e. intracellular Aβ levels between male and female APP23 mice (Fig 2D). Analysis of pro- and anti-inflammatory cytokines (IFNγ, IL-10, IL-1β, IL-2, IL-4, IL-5, IL-6, TNF-α, CXCL1) revealed increased levels of IL-4 in the plasma, IL-10 in both plasma and brain as well as TNF-α and CXCL1 in brain homogenates of female APP23 mice, compared to male mice (Fig EV2A–I). We also noted a positive correlation between CXCL1 levels in the brain and both soluble and insoluble Aβ$_{1-40}$ levels (Fig EV1B). In summary, male and female APP23 mice show distinct differences in plaque accumulation at an age of 21 months. In both biochemical and histological analyses, female APP23 mice had twice the amount of Aβ accumulation and plaque load, and Aβ aggregates are more insoluble and more abundant than those in age-matched male APP23 mice. We also identified an Aβ burden-dependent increase in cortical astrocytes in female APP23 mice while microgliosis and BACE1-positive dystrophic neurites were unchanged between genders. Moreover, brain and plasma cytokine levels were regulated differently in male versus female APP23 mice. Due to the observed gender differences in Aβ deposition and associated pathology, the impact of IL12p40 deficiency on pathological outcomes in this mouse model was assessed separately in female and male APP23 mice.

## Microglial IL12p40 is increased in APP23 mice

To examine whether the IL12p40 subunit also is a relevant interventional immune target in the APP23 mouse model, we examined IL12p40 expression in aged APP23 mice. Firstly, we could validate our previous findings [18], namely that IL12p40 (Il12b) gene expression in the brain was specific to microglia irrespective of gender (Fig 3A). On protein level, APP23 mice showed a ~45% increase in IL12p40 compared to age-matched wild-type (WT) littermates that were not influenced by the gender of the mice (Fig 3B).

To further understand the role of IL12p40 on disease pathophysiology in this mouse model, we crossed APP23 mice to IL12p40$^{-/-}$ knock out mice [49] (APP23p40$^{-/-}$) and investigated whether a lack of the IL12p40 subunit influences plaque pathology as it did in the APPPS1 model [18]. We confirmed by ELISA analysis that APP23p40$^{-/-}$ mice lack IL12p40 expression irrespective of the gender of the mice (Fig 3B).

## Male APP23p40$^{-/-}$ mice exhibit reduced Aβ deposits compared to APP23 mice

To investigate the effect of IL12p40 deficiency, brain tissues of male and female APP23 and APP23p40$^{-/-}$ mice were analysed for Aβ levels, the abundance of Aβ aggregates, surrogate markers of neuritic dystrophy, astrogliosis, plaque-associated microglia, the levels of pro- and anti-inflammatory cytokines as well as Aβ processing enzymes using histological and biochemical methods.

In male mice, biochemical analysis of Aβ$_{1-40}$ levels in TBS, Triton-X and SDS-soluble protein fractions did not reveal any differences between the APP23 and APP23p40$^{-/-}$ genotypes (Fig 4A). Additionally, Aβ$_{1-42}$ concentration was not influenced by IL12p40 deficiency (Appendix Fig S2A), as was the presence of Aβ aggregates as measured by native filter test (Fig 4E; Appendix Fig S3). Histological analyses of brain sections, however, did reveal a strong reduction in the area covered of both diffuse and compact plaques. Here, genetic deficiency of IL12p40 led to a 58% decrease in the cortical area covered by 4G8-positive plaques (Fig 4B) and to a 42% reduction in the area covered by pFTAA-positive plaques (Fig 4C). Similarly, Congo Red-positive "core" plaques were reduced by 52% (Fig 4D). The presence of filtered Aβ aggregates (Fig 4E), the total ratio of diffuse versus core plaques (Appendix Fig S2C) and expression levels of APP, APP-cleaving protein BACE1 and Aβ degrading enzymes neprilysin and insulin-degrading enzyme (IDE) were not influenced by IL12p40 deficiency (Fig EV3A and B), similar to what has been reported for APPPS1 mice lacking IL12p40 [18]. Deficiency of IL12p40 also did not affect plaque-associated BACE1 immunoreactivity or astrocyte numbers in male APP23 mice (Fig 5A and B), which also showed no alteration in the number of plaque-associated microglia and of Clec7a-positive activated microglia (Fig 5C) or Aβ uptake (Fig 5D). While most pro- and anti-inflammatory cytokines assessed by us were not altered in brain homogenates of male APP23 and APP23p40$^{-/-}$ mice, we noted a threefold reduction in IFNγ levels in plasma samples of male mice lacking IL12p40 (Fig EV3C–K).

## Female APP23p40$^{-/-}$ mice show a reduction in soluble and insoluble Aβ$_{1-40}$ compared to APP23 mice

In contrast to male mice, IL12p40 deficiency in female APP23 mice had substantial effects on Aβ levels. In the TBS and Triton-X soluble protein fractions, a 38% and a 45% decrease in Aβ$_{1-40}$ levels could be detected in female APP23 versus APP23p40$^{-/-}$ mice (Fig 6A). No effect was seen on SDS-soluble Aβ (Fig 6A) and aggregated Aβ

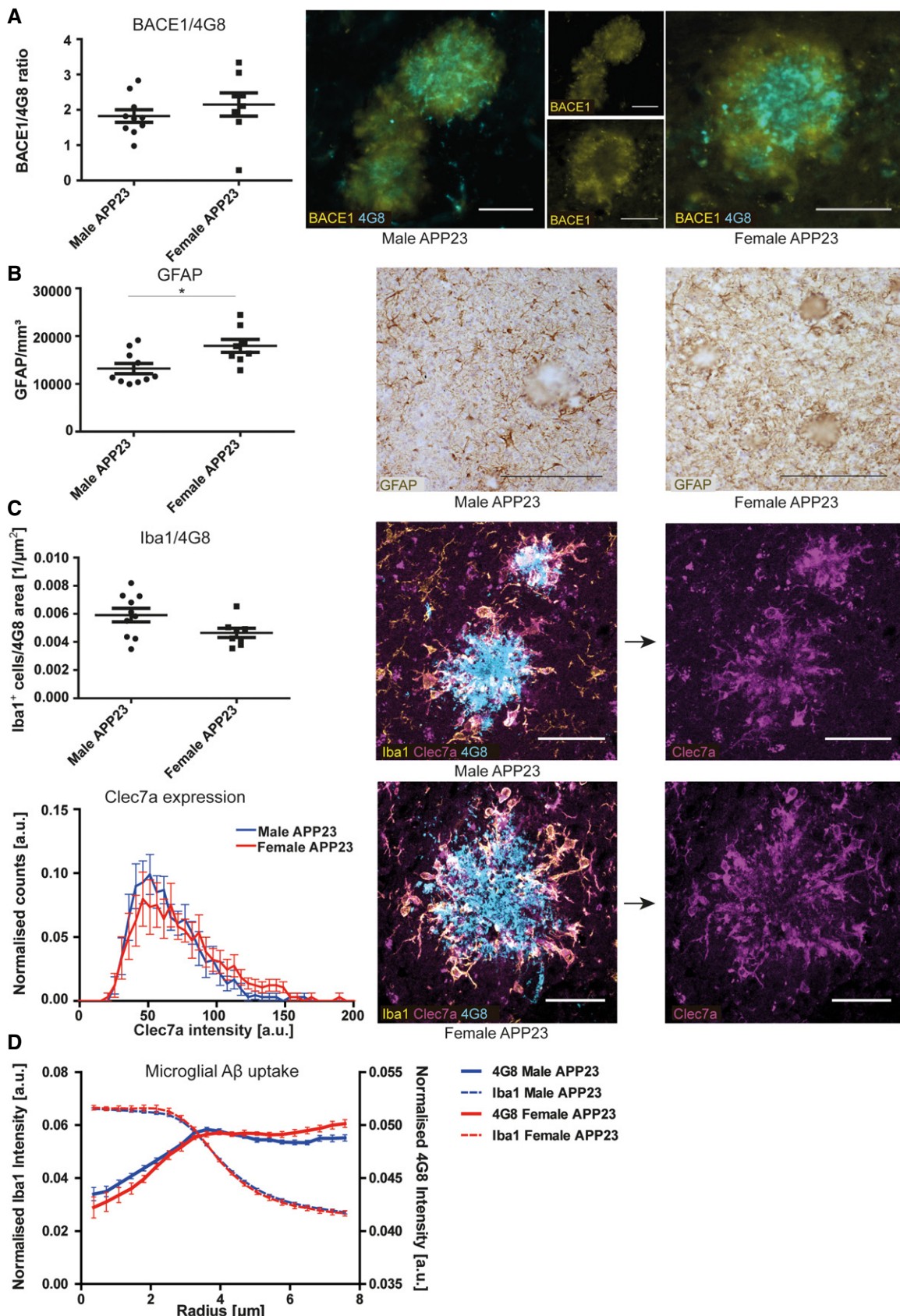

Figure 2.

◄

**Figure 2. Female APP23 mice have higher astrocyte numbers than male mice.**

A  Histological analysis of plaque-associated BACE1 immunoreactivity in male (n = 10) and female (n = 8) APP23 mice. BACE1 area covered was normalised to 4G8-positive area covered of the same image (left). Right: representative images, scale bar = 50 μm. Mean ± SEM, statistical analysis: two-tailed unpaired t-test, P = 0.3724.

B  Stereological quantification of the number of cortical GFAP-positive astrocytes in male (n = 10) and female (n = 8) APP23 mice (left). Right: representative images of GFAP staining, scale bar = 200 μm. Mean ± SEM, statistical analysis: two-tailed unpaired t-test, *P = 0.0134.

C  Quantification of activated microglia within 30 μm from plaque borders. Top: numbers of Iba1-positive microglia were normalised to the size of the nearest 4G8-positive plaque and quantified in male (n = 10) and female (n = 8) APP23 mice. Mean ± SEM, statistical analysis: two-tailed unpaired t-test, P = 0.0576. Bottom: histogram representing Clec7a staining intensity within plaque-associated Iba1-positive microglia in male (n = 10) and female (n = 8) APP23 mice. Mean ± SEM, statistical analysis: two-tailed unpaired t-test with Bonferroni correction for each single bin, P = N.S.. Right: representative images, scale bar = 40 μm.

D  Radial intensity profiles of Iba1 and 4G8 around the centre of the nucleus of plaque-associated Iba1-positive microglia in male (n = 10) and female (n = 8) APP23 mice. Iba1 intensity declines until a radius of ~6 μm, marking the cell periphery. 4G8 intensity peaks inside the cell (~4 μm), but stays high outside the cell. This is very likely due to the close proximity to 4G8-positive plaques. Mean ± SEM, statistical analysis: two-tailed unpaired t-test with Bonferroni correction for the number of binned radii shows no significant difference between both 4G8 traces.

Source data are available online for this figure.

as measured by filter assays (Fig 6E; Appendix Fig S3). As in male mice, Aβ$_{1-42}$ levels were not affected by a lack of IL12p40 (Appendix Fig S2B) as were the expression levels of APP, BACE1, Neprilysin and IDE (Fig EV4A and B).

To confirm whether these changes also affected the amount of deposited Aβ, brain sections of female APP23 and APP23p40$^{-/-}$ mice were stained with 4G8, pFTAA and Congo Red. The percentage of the cortical area covered by 4G8-, pFTAA- and Congo Red-positive plaques was unchanged in female APP23 mice lacking or harbouring IL12p40 (Fig 6B–D), contrary to the findings in male mice. The ratio of diffuse versus core plaques also was not influenced by a lack of IL12p40 (Appendix Fig S2D). Plaque-associated BACE1 immunoreactivity, cortical astrocyte number, the number of activated plaque-associated microglia and microglial Aβ uptake were all unchanged in female APP23p40$^{-/-}$ mice (Fig 7A–D).

Analysis of pro- and anti-inflammatory cytokines (IFNγ, IL-10, IL-1β, IL-2, IL-4, IL-5, IL-6, TNF-α, CXCL1) in plasma showed reduced expression of IL-5 and IL-6 as well as an increased expression of IL-1β and CXCL1 in female APP23p40$^{-/-}$ mice when compared to APP23 mice. In brain homogenates, a twofold decrease in CXCL1 protein levels was noted in APP23p40$^{-/-}$ mice (Fig EV4C–K).

## Discussion

Using the Aβ-producing APP23 mouse model of AD-like pathology, we identified specific gender differences in plaque accumulation, amyloid composition and aggregation characteristics, astrogliosis as well as brain and plasma cytokine levels. We further show that the deletion of the IL12p40 subunit, which is the essential component of the cytokines IL-12 and IL-23, differentially affects pathology in age-matched male and female mice.

Given that male and female APP23 mice are known to show varying levels of pathology and behavioural deficits, most studies using APP23 mice analyse male and female animals independently [31,32]. Previously observed sex differences in spatial learning paradigms may be explained by hormonal variances [50], yet an influence of hormones upon plaque pathology has not been described in

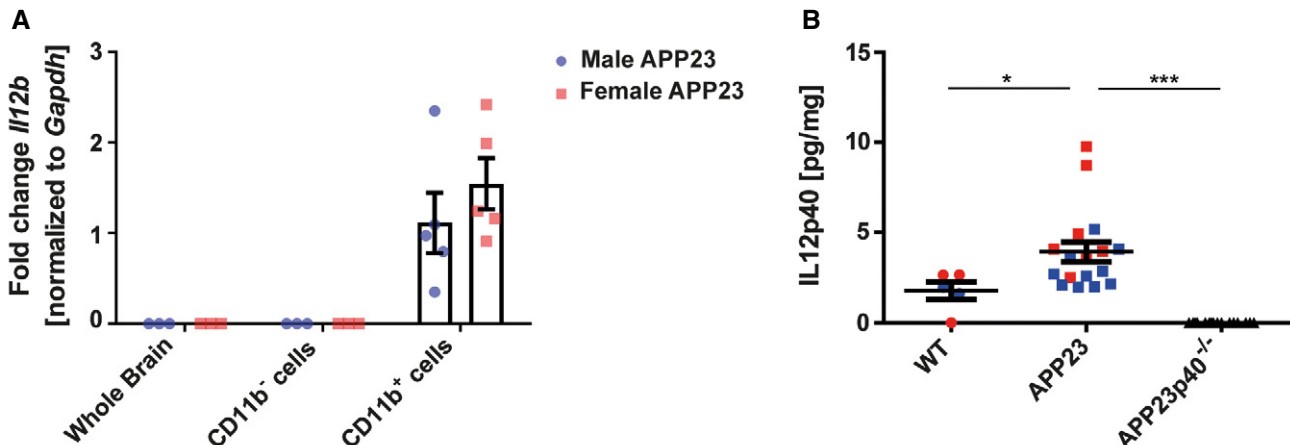

**Figure 3. APP23 mice have increased microglia-specific Il12b/IL12p40 levels in the brain.**

A  Gene expression analysis of IL12b in whole brain (n = 3 per gender, Il12b undetected), Cd11b-negative non-microglial cells (n = 3 per gender, Il12b undetected) and CD11b-positive microglia (n = 5 per gender, P = 0.3540) in male and female APP23 mice. Gapdh expression was used as an internal reference gene. Mean ± SEM, statistical analysis: two-tailed unpaired t-test for each fraction.

B  ELISA measurements of the IL12p40 concentration in the TBS-soluble protein fraction derived from wild-type (WT) (n = 5), APP23 (n = 17) and APP23p40$^{-/-}$ (n = 16) mice. Male mice are depicted by blue squares, while female mice are shown as red squares. Mean ± SEM, statistical analysis: one-way ANOVA, Tukey post hoc test, *P = 0.0276, ***P = < 0.0001.

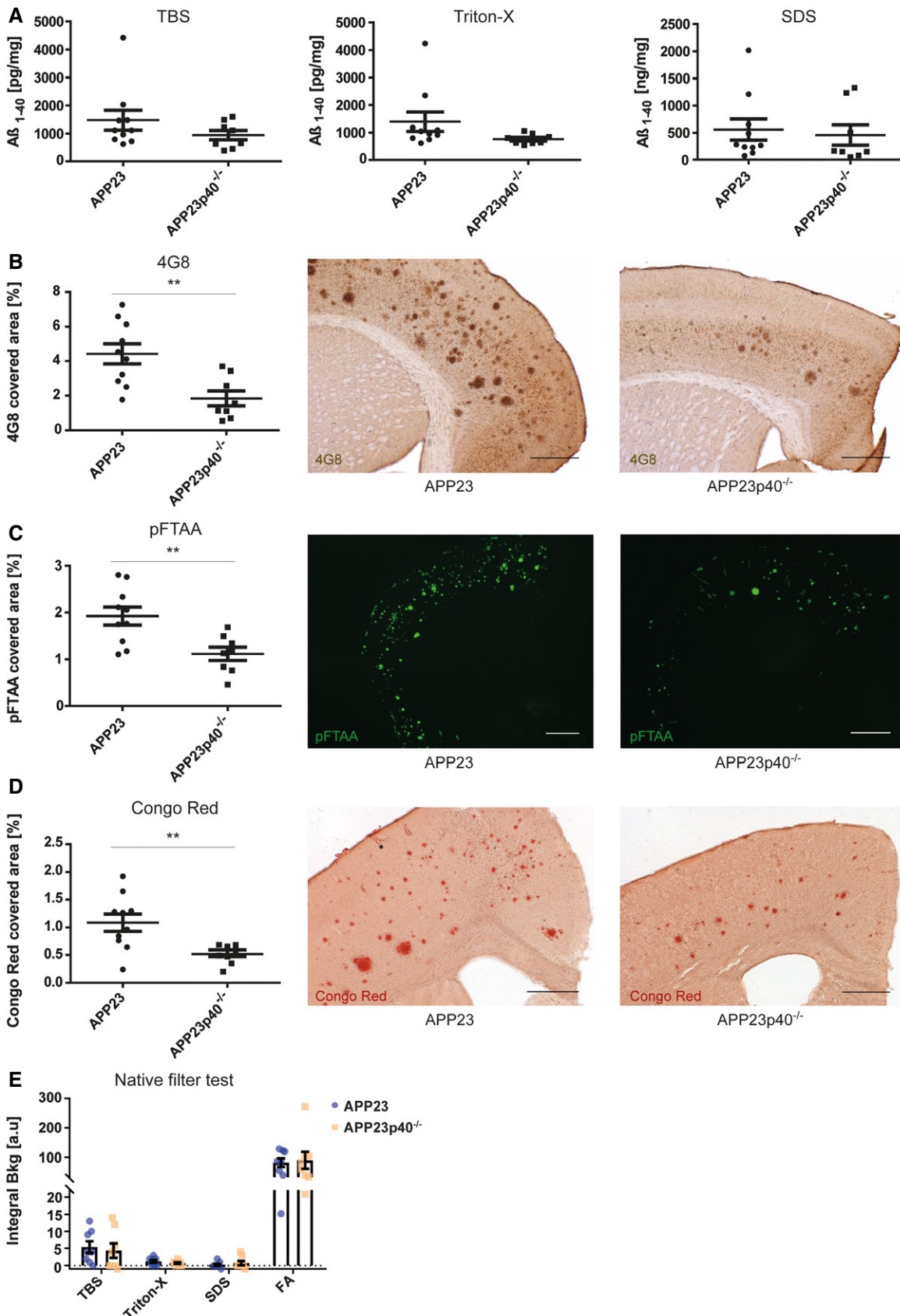

Figure 4.

**Figure 4. In male APP23 mice, IL12p40 deficiency reduces Aβ plaque deposition but not does not affect biochemical characteristics of Aβ.**

A   Mesoscale analysis for the $Aβ_{1-40}$ protein in the TBS ($P = 0.2298$), Triton-X ($P = 0.1329$) and SDS ($P = 0.7184$) fractions of brain homogenates from male APP23 ($n = 10$) and APP23p40$^{-/-}$ ($n = 8$) mice. Total protein concentration of each sample was used as an internal reference. Mean ± SEM, statistical analysis: two-tailed unpaired $t$-test.

B   Stereological analysis of cortical area covered by 4G8-positive plaques (left) and representative images of 4G8-staining in APP23 ($n = 10$) and APP23p40$^{-/-}$ ($n = 8$) mice (right), scale bar = 500 μm. Mean ± SEM, statistical analysis: two-tailed unpaired $t$-test, **$P = 0.0039$.

C   Fluorescence intensity-based analysis of pFTAA-positive area covered in the cortex of APP23 ($n = 10$) and APP23p40$^{-/-}$ ($n = 8$) mice (left) and representative images for each genotype (right), scale bar = 1 mm. Mean ± SEM, statistical analysis: two-tailed unpaired $t$-test, **$P = 0.0051$.

D   Stereological analysis of cortical area covered by Congo Red-positive plaques in APP23 ($n = 10$) and APP23p40$^{-/-}$ ($n = 8$) mice (left) and representative images (right), scale bar = 500 μm. Mean ± SEM, statistical analysis: two-tailed unpaired $t$-test, **$P = 0.0070$.

E   Native filter assay analysis of TBS ($P = 0.7124$), Triton-X ($P = 0.3170$), SDS ($P = 0.4833$) and formic acid (FA) ($P = 0.8144$) fractions from APP23 ($n = 8$) and APP23p40$^{-/-}$ ($n = 8$) mouse brain homogenates. Mean ± SEM, statistical analysis: two-tailed unpaired $t$-test between the same fractions.

this mouse model to date. Since none of the studies assessing plaque pathology in APP23 mice undertook thorough comparative analyses of aged male and female mice [31,50–52], and as recent reports highlight differences in the innate immune response between males and females including differences in male and female microglia [25–28], we undertook gender-specific analyses of various pathological readouts where we observed robust differences between male and female mice at late stages of pathology (21 months). Female APP23 mice showed a twofold higher pathology in both biochemical and histological analyses of Aβ accumulation compared to male mice, indicating faster disease progression. Furthermore, Aβ aggregation properties differed between genders since in female mice increased amounts of Aβ aggregates were found in SDS- and FA-extracted protein fractions. In male mice, aggregates were mainly found in TBS- and FA-soluble fractions. Previous studies have shown that the vast majority of Aβ plaques in APP23 mice in an age range of 14–24 months are densely aggregated core plaques and that diffuse plaques only appear in very old mice [29,39,53]. Contrary to these reports, diffuse Aβ plaques, as visualised by immunohistochemistry, constituted the majority of plaques in our analyses of 21-month-old male and female mice, their levels being four times as high as core plaques. We also did not observe any gender-specific differences in plaque-associated BACE1 immunoreactivity, a surrogate marker of neuritic dystrophy [40–43]. Additionally, female APP23 mice, in accordance with the increase in plaque burden, also showed higher numbers of cortical astrocytes and increased levels of CXCL1 (also known as KC/CRO or GRO1) in the brain, which positively correlated with $Aβ_{1-40}$ pathology. It is of note that in a mouse model of multiple sclerosis, astrocyte-specific induction of CXCL1 augmented disease progression via recruitment of neutrophils [54], while in an

AD-like mouse model, blocking the entry of neutrophils into the brain was shown to have a beneficial effect upon pathogenesis [55]. CXCL1, as one of the differentially regulated cytokines in male versus female APP23 mice lacking or harbouring IL12p40, may thus not only present a possible (non-exclusive) explanation for the gender-specific differences in AD pathology, but may also qualify as an interesting target to study in AD pathogenesis. While the cytokine signatures in brain and plasma of male and female APP23 mice seem to differ, we could not observe any gender-specific differences in the number of plaque-associated microglia, their expression of Clec7a, a marker of activated microglia in various disease contexts [47,48] and Aβ uptake, suggesting that altered microglial functions are not the cause of gender-specific pathogenesis in this mouse model.

We previously reported that genetic deletion or pharmacological blockage of the pro-inflammatory IL12p40 in the APPPS1 mouse model led to a marked decrease in plaque pathology at both early and late stages of Aβ deposition (4 and 8 months, respectively) as well as a reduction in cognitive deficits [18]. This study also found that the IL12p40 subunit was expressed by microglia, describing for the first time a role of IL-12/IL-23 signalling in AD carried out by glial cells in the CNS. Given that the APPPS1 mouse model is characterised by a rapid accumulation of Aβ deposits, it may not fully represent the rather slow Aβ accumulation and disease progression that typically is described for sporadic human AD. Our data using the APP23 AD-like mouse model now show that a lack of IL12p40 similarly leads to a reduction in Aβ burden in a mouse model with a rather slow disease course. We therefore provide further evidence of the involvement of IL-12 and/or IL-23 signalling in AD pathogenesis, which also strengthens the hypothesis that the blockage of certain pro-inflammatory factors secreted by glia can have beneficial

**Figure 5. Lack of IL12p40 does not affect plaque-associated BACE1 immunoreactivity or glial properties in male APP23 mice.**

A   Histological analysis of plaque-associated BACE1 immunoreactivity in male APP23 ($n = 10$) and APP23p40$^{-/-}$ ($n = 8$) mice. BACE1 area covered was normalised to 4G8-positive area covered of the same image (left). Right: representative images, scale bar = 50 μm. Mean ± SEM, statistical analysis: two-tailed unpaired $t$-test, $P = 0.1780$.

B   Stereological quantification of the number of cortical GFAP-positive astrocytes in male APP23 ($n = 10$) and APP23p40$^{-/-}$ ($n = 8$) mice (left). Right: representative images of GFAP staining, scale bar = 200 μm. Mean ± SEM, statistical analysis: two-tailed unpaired $t$-test, $P = 0.3148$.

C   Quantification of activated microglia within 30 μm from plaque borders. Top: numbers of Iba1-positive microglia were normalised to the size of the nearest 4G8-positive plaque and quantified in male APP23 ($n = 10$) and APP23p40$^{-/-}$ ($n = 8$) mice. Mean ± SEM, statistical analysis: two-tailed unpaired $t$-test, $P = 0.1925$. Bottom: histogram representing Clec7a staining intensity within plaque-associated Iba1-positive microglia in male APP23 ($n = 10$) and APP23p40$^{-/-}$ ($n = 8$) mice. Mean ± SEM, statistical analysis: two-tailed unpaired $t$-test with Bonferroni correction for each single bin, $P = N.S.$ Right: representative images, scale bar = 40 μm.

D   Radial intensity profiles of Iba1 and 4G8 around the centre of the nucleus of plaque-associated Iba1-positive microglia in male APP23 ($n = 10$) and APP23p40$^{-/-}$ ($n = 8$) mice. Iba1 intensity declines until a radius of ~6 μm, marking the cell periphery. 4G8 intensity peaks inside the cell (~4 μm), but stays high outside the cell. This is very likely due to the close proximity to 4G8-positive plaques. Mean ± SEM, statistical analysis: two-tailed unpaired $t$-test with Bonferroni correction for the number of binned radii shows no significant difference between both 4G8 traces.

Source data are available online for this figure.

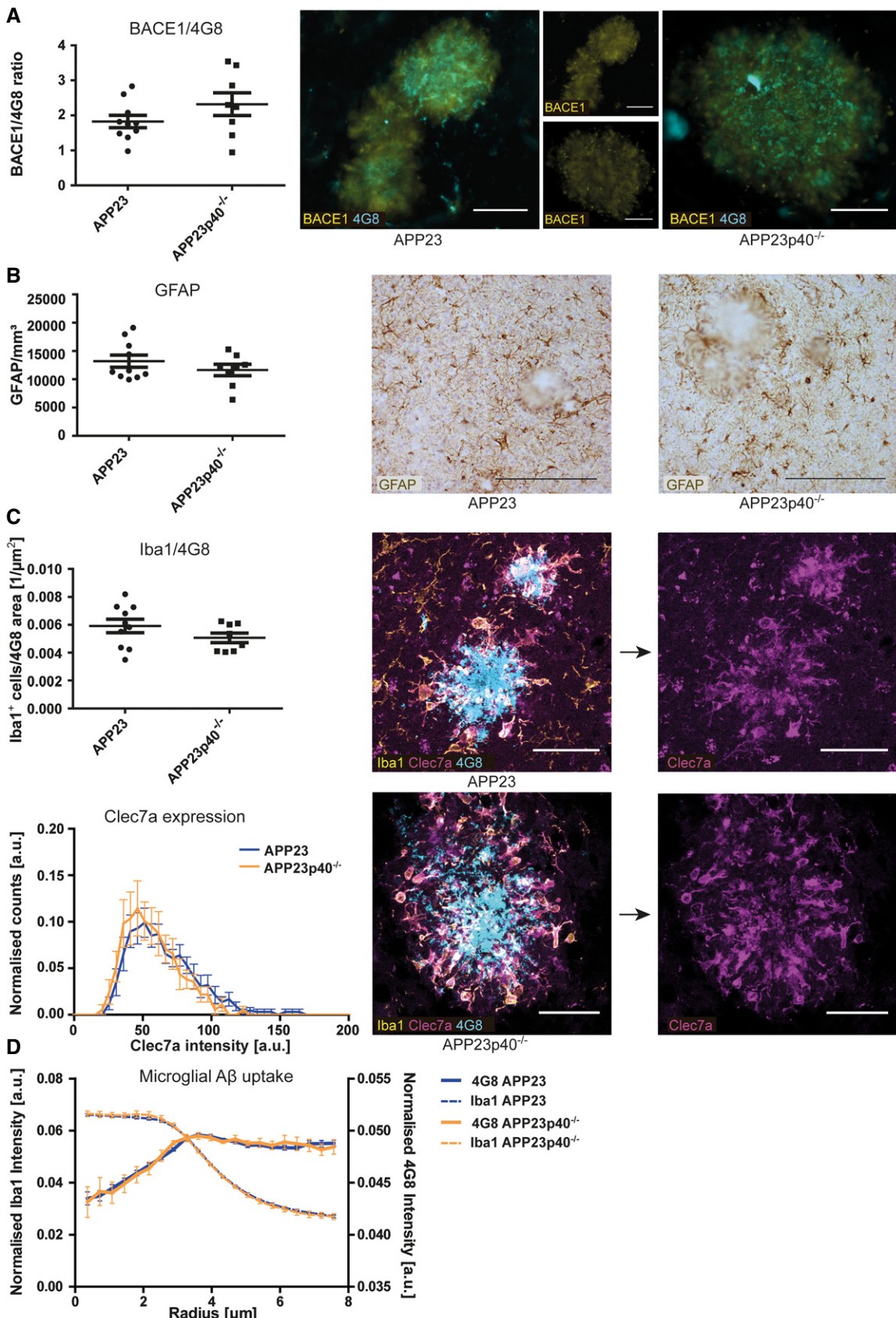

**Figure 5.**

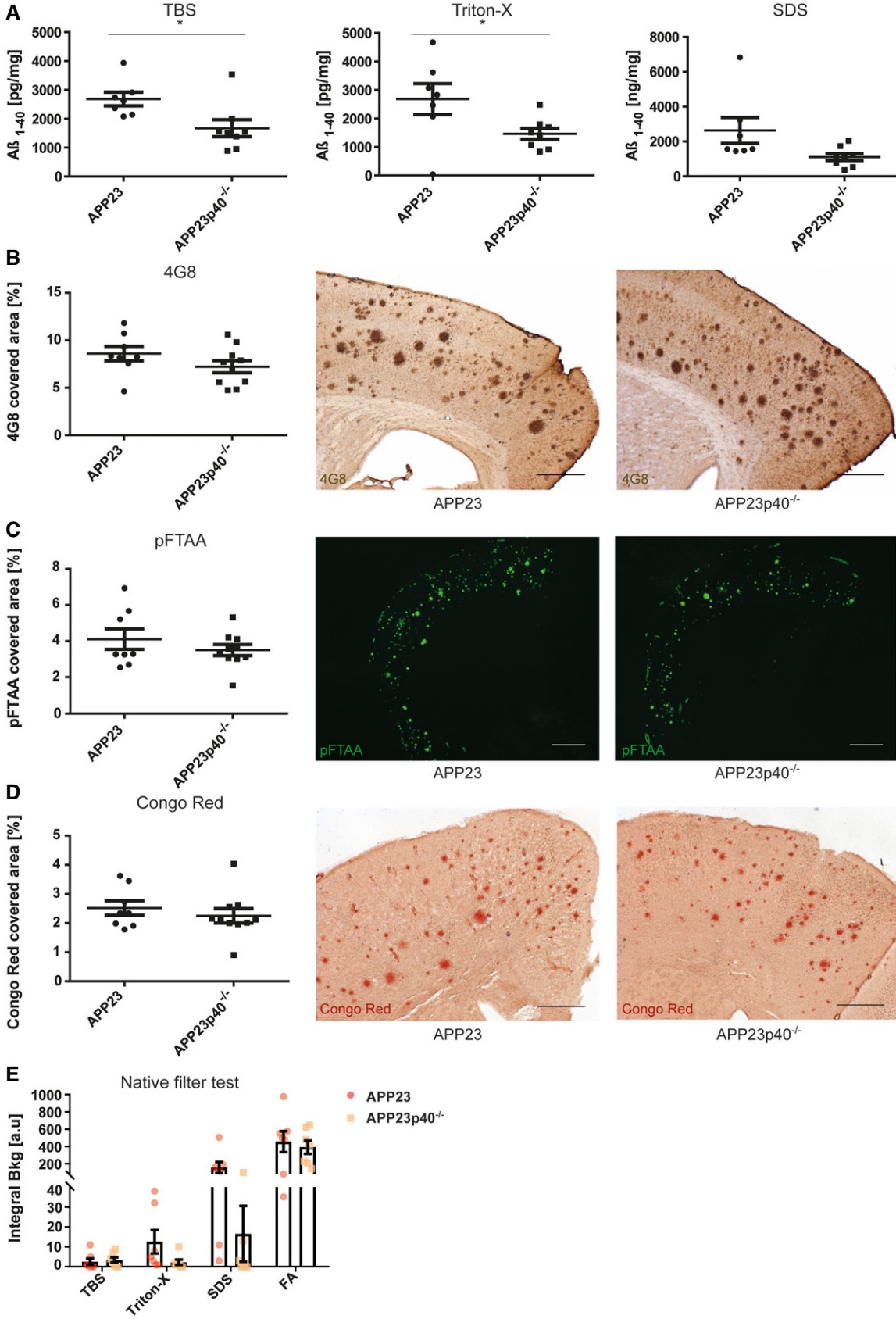

Figure 6.

◄

Figure 6. In female APP23 mice, IL12p40 deletion leads to a reduction in soluble Aβ$_{1-40}$ but not Aβ plaque load.

A Mesoscale analysis for the Aβ$_{1-40}$ protein in the TBS (*$P$ = 0.0208), Triton-X (*$P$ = 0.0440) and SDS ($P$ = 0.0540) fractions of brain homogenates from APP23 ($n$ = 7) and APP23p40$^{-/-}$ ($n$ = 8) mice. Total protein concentration of each sample was used as an internal reference. Mean ± SEM, statistical analysis: two-tailed unpaired $t$-test.

B Stereological analysis of cortical area covered by 4G8-positive plaques (left) and representative images of 4G8-staining in APP23 ($n$ = 8) and APP23p40$^{-/-}$ ($n$ = 10) mice (right), scale bar = 500 μm. Mean ± SEM, statistical analysis: two-tailed unpaired $t$-test, $P$ = 0.1831.

C Fluorescence intensity-based analysis of pFTAA-positive area covered in the cortex of APP23 ($n$ = 8) and APP23p40$^{-/-}$ ($n$ = 10) mice (left) and representative images for each genotype (right), scale bar = 1 mm. Mean ± SEM, statistical analysis: two-tailed unpaired $t$-test, $P$ = 0.3406.

D Stereological analysis of cortical area covered by Congo Red-positive plaques in APP23 ($n$ = 8) and APP23p40$^{-/-}$ ($n$ = 10) mice (left) and representative images (right), scale bar = 500 μm. Mean ± SEM, statistical analysis: two-tailed unpaired $t$-test, $P$ = 0.4542.

E Native filter assay analysis of TBS ($P$ = 0.6823), Triton-X ($P$ = 0.1146), SDS ($P$ = 0.0508) and formic acid (FA) ($P$ = 0.6603) fractions from APP23 ($n$ = 7) and APP23p40$^{-/-}$ ($n$ = 7) mouse brain homogenates. Mean ± SEM, statistical analysis: two-tailed unpaired $t$-tests between the same fractions.

effects upon Aβ pathology. The importance of IL-12/IL-23 signalling in AD is supported further by other studies demonstrating that targeting IL12p40 via small interfering RNA in the SAMP8 AD-like mouse model of accelerated ageing induced a reduction in cerebral Aβ as well as reduced neuronal loss and cognitive function [33]. Analysis of genetic data within the Han Chinese population also indicated specific polymorphisms in the IL-12/IL-23 pathway as risk factors for late-onset AD [56,57]. Additionally, increased levels of IL-23 and/or IL-12 were found in serum and plasma [19,58] and a correlation was made between IL12p40 CSF levels and cognitive performance in AD patients [18]. Given that IL-12/IL-23 has been shown to be regulated in MCI and AD subjects [18–20] and biologicals that inhibit IL-12 and/or IL-23 have already been approved by the US Food and Drug Administration (FDA) for other diseases such as psoriasis and Crohn's disease, the immediate suitability for repurposing existing drugs targeting these innate immune molecules in a first clinical AD trial is obvious.

In addition to previous data assessing the effect of a lack of IL12p40 on Aβ plaque burden in AD-like mice, we noted differential effects of IL-12/IL-23 deficiency in age-matched female and male APP23 mice. Compared to APP23 mice, male APP23p40$^{-/-}$ mice showed a significant reduction in diffuse and core plaques when histologically assessing cortical plaque burden, while Aβ$_{1-40}$ levels and Aβ aggregation properties were not altered. Contrary to male mice, female APP23p40$^{-/-}$ mice did have reduced levels of soluble Aβ$_{1-40}$ when compared to APP23 mice, though cortical plaque burden appeared to not be affected. In both male and female mice, the observed changes in Aβ pathology upon IL12p40 deficiency were not mediated by differential APP expression or Aβ processing. Interestingly, despite the differences in Aβ pathology, BACE1-positive

dystrophic neurites, the number of cortical astrocytes or plaque-associated microglia were not affected upon IL12p40 deletion, including the expression of microglial Clec7a and Aβ uptake. These observations suggest that microglial IL12p40 does not seem to excerpt its detrimental effect upon Aβ pathology by modulating microglial functions. Alternatively, an indirect effect of IL12p40-mediated intercellular signalling could take place given that the IL12p40 receptor was found to be expressed on non-microglial cells in an AD model [18]. Since IL12p40 deficiency does not affect Aβ processing, it could act upon other cell types by restoring cellular metabolism and thus intracellular degradation of Aβ or by modulating peripheral cells that might affect Aβ deposition such as neutrophils via CXCL1 [55]. The gender-specific effects of IL12p40 deletion upon Aβ pathology could also be explained by the underlying pathological differences between male and female APP23 mice at the age of 21 months. Pathology in female APP23 mice could already be so advanced that potential effects of lack of IL12p40 are overshadowed by the abundance of Aβ deposits. In male APP23 mice, on the other hand, fewer Aβ aggregates at a given stage are present which is why effects of IL12p40 deficiency on plaque accumulation are still observable.

In summary, we show that genetic ablation of the IL-23/IL-12 p40 subunit has a different effect on plaque and cellular pathology in age-matched male and female APP23 mice, a mouse model of slow Aβ accumulation with gender-specific temporal pathogenesis. While in female APP23 mice, deletion of IL-12/IL-23 signalling specifically decreased soluble Aβ$_{1-40}$ levels, the pathology in male mice was characterised by a reduction in cortical Aβ plaque load. Our results provide important evidence on the role of IL-12 and IL-23 signalling in a mouse model of amyloid deposition, which adds to data suggesting a detrimental effect of this signalling cascade

Figure 7. Lack of IL12p40 but does not affect plaque-associated BACE1 immunoreactivity or gliosis in female APP23 mice.

A Histological analysis of plaque-associated BACE1 immunoreactivity in female APP23 ($n$ = 8) and APP23p40$^{-/-}$ ($n$ = 10) mice. BACE1 area covered was normalised to 4G8-positive area covered of the same image (left). Right: representative images, scale bar = 50 μm. Mean ± SEM, statistical analysis: two-tailed unpaired $t$-test, $P$ = 0.5402.

B Stereological quantification of the number of cortical GFAP-positive astrocytes in female APP23 ($n$ = 8) and APP23p40$^{-/-}$ ($n$ = 8) mice (left). Right: representative images of GFAP staining, scale bar = 200 μm. Mean ± SEM, statistical analysis: two-tailed unpaired $t$-test, $P$ = 0.1579.

C Quantification of activated microglia within 30 μm from plaque borders. Top: numbers of Iba1-positive microglia were normalised to the size of the nearest 4G8-positive plaque and quantified in female APP23 ($n$ = 8) and APP23p40$^{-/-}$ ($n$ = 10) mice. Mean ± SEM, statistical analysis: two-tailed unpaired $t$-test, $P$ = 0.8240. Bottom: Histogram representing Clec7a staining intensity within plaque-associated Iba1-positive microglia in female APP23 ($n$ = 8) and APP23p40$^{-/-}$ ($n$ = 10) mice. Mean ± SEM, statistical analysis: two-tailed unpaired $t$-test with Bonferroni correction for each single bin, $P$ = N.S.. Right: representative images, scale bar = 40 μm.

D Radial intensity profiles of Iba1 and 4G8 around the centre of the nucleus of plaque-associated Iba1-positive microglia in female APP23 ($n$ = 8) and APP23p40$^{-/-}$ ($n$ = 10) mice. Iba1 intensity declines until a radius of ~6 μm, marking the cell periphery. 4G8 intensity peaks inside the cell (~4 μm), but stays high outside the cell. This is very likely due to the close proximity to 4G8-positive plaques. Mean ± SEM, statistical analysis: two-tailed unpaired $t$-test with Bonferroni correction for the number of binned radii shows no significant difference between both 4G8 traces.

Source data are available online for this figure.

▶

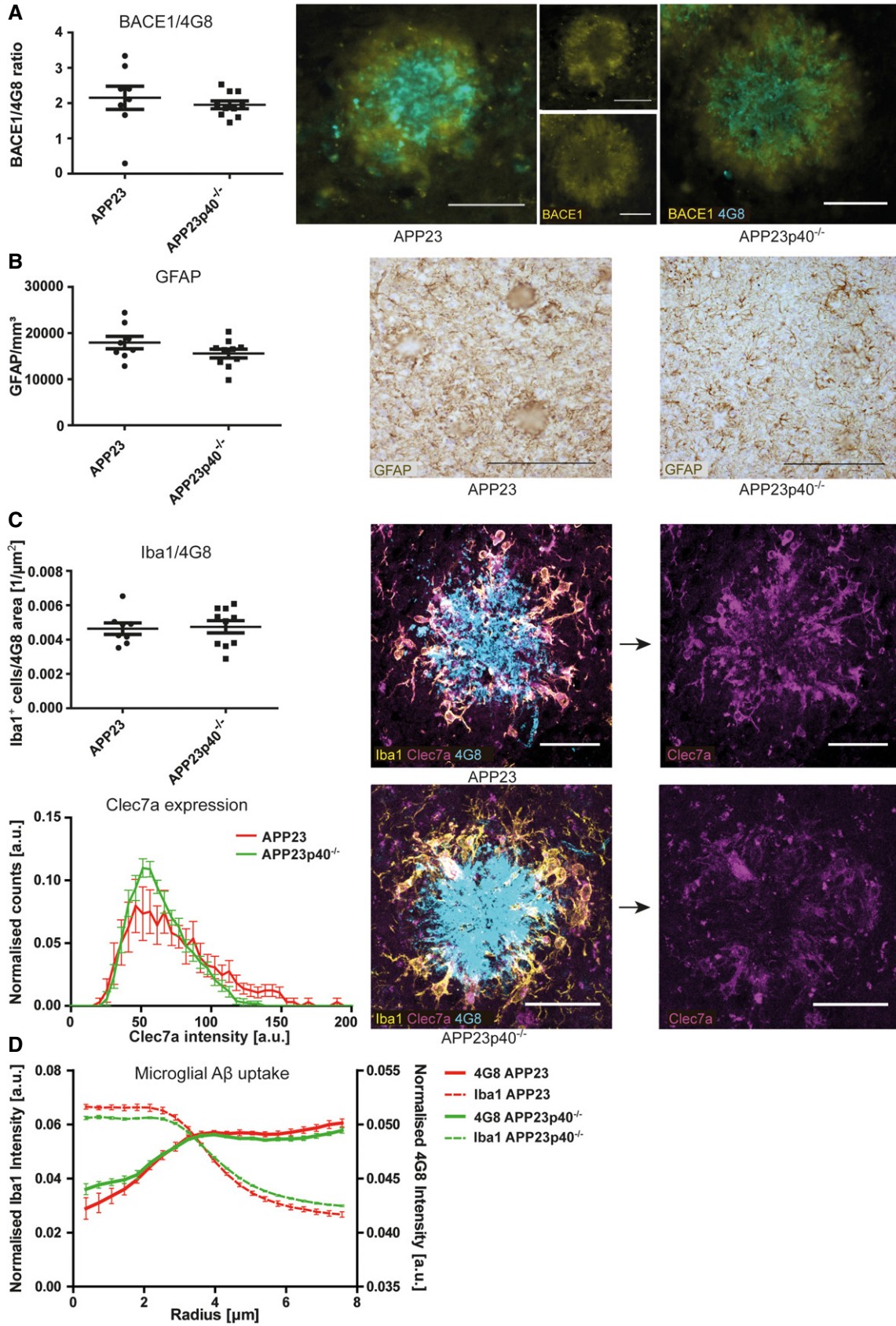

Figure 7.

[18–20,33,56–58]. While future research aimed at successfully targeting IL12p40 in AD implies the need to dissect its downstream mechanisms and to identify whether it is IL-12 or IL-23 specifically that influences AD pathogenesis, it will be equally interesting to address whether this signalling pathway also plays a gender-specific role in other CNS proteinopathies such as Parkinson's disease or tau-driven pathologies.

# Materials and Methods

### Animals

We crossed heterozygous APP23$^{+/-}$ mice (Tg(Thy1-APPKM670/671NL)23) [29], termed APP23 throughout this manuscript, to mice lacking the IL12p40 gene *Il12b*, termed *Il12p40*$^{-/-}$ mice [47], generating APP23p40$^{-/-}$ mice and APP23 littermate controls. Cohorts of male and female APP23 mice were compared to each other and used as control groups in comparison to APP23p40$^{-/-}$ littermates. Thus, data points of male and female APP23 mice shown in Figs 1A–E, 2A–D, EV2A–I and S1A–B served also as references in Figs 4A–E, 5A–D, EV3C–K and S2A and C analysing male APP23 mice as well as in Figs 6A–E, 7A–D, EV4C–K and S2B and D analysing female APP23 mice. Mice were group-housed under specific pathogen-free conditions on a 12-h light/dark cycle; food and water were provided *ad libidum*. We did not observe any differences in mortality between male and female APP23 and APP23p40$^{-/-}$ mice. All animal experiments were performed in accordance with the national animal protection guidelines approved by the regional offices for health and social services in Berlin (LaGeSo, licence number O 0132/09).

### Tissue processing

Transgenic APP23 and APP23p40$^{-/-}$ mice and littermate controls were aged to 21 months. For tissue collection, mice were deeply anaesthetised and transcardially perfused with PBS. Venous blood was collected from the right atrium into EDTA-coated tubes. After centrifugation, the plasma supernatant was collected, snap-frozen in liquid nitrogen and stored at −80°C. Plasma samples could not be collected for all experimental animals. Brains were rapidly removed from the skull and divided into half sagittally, and the cerebellum was removed. One hemisphere was snap-frozen in liquid nitrogen and stored at −80°C until further processing for biochemical analysis, and the other hemisphere was fixed in 4% paraformaldehyde over night at 4°C. Subsequently, the hemisphere was immersed in 30% sucrose for at least 24 h until sectioning for immunohistochemical analysis. For some animals, the fresh frozen hemispheres were not available for further biochemical analyses. For RNA analysis, tissue from male and female APP23 mice aged 648–764 days was processed as described below.

### RNA isolation of brain fractions

For RNA analysis, the left hemisphere was used to isolate microglia and the microglia-negative fraction from fresh tissue while the right hemisphere was snap-frozen in liquid nitrogen to generate RNA from whole brain. Microglia were isolated using magnetic-activated cell sorting (MACS) according to manufacturer's protocol. In brief, tissue was dissociated using the Neural Tissue Dissociation Kit (P) (Miltenyi Biotec, 130-092-628) on a gentleMACS Octo Dissociator with Heaters (Miltenyi Biotec, 130-096-427) and the resulting single-cell suspension labelled with CD11b MicroBeads (Miltenyi Biotec, 130-093-634) and passed through LS columns (Miltenyi Biotec, 130-042-401) to positively select for microglia. The CD11b-negative flow-through was also collected as the microglia-negative brain fraction. Both cell fractions were pelleted via centrifugation, snap-frozen and stored at −80°C until further use. For whole brain RNA, the frozen right hemisphere was homogenised in RLT buffer (RNeasy Mini Kit, Qiagen, 74106) using M tubes (Miltenyi Biotec, 130-093-236) on the gentleMACS Octo Dissociator with Heaters (Miltenyi Biotec, 130-096-427) before continuing with the downstream RNA isolation protocol. For RNA isolation, the RNeasy Mini Kit (Qiagen, 74106) was used and cDNA was generated using the High-Capacity cDNA Reverse Transcription Kit (Thermo Fisher, 4368813) according to manufacturer's protocols.

### Quantitative real-time PCR

Gene expression analysis was performed on 12 ng cDNA per reaction using the TaqMan Fast Universal Master Mix (Applied Biosystems, 4364103) and TaqMan primers for *Il12b* (Thermo Fisher, Mm00434174_m1) and *Gapdh* (Thermo Fisher, Mm99999915_g1). Quantitative PCR analysis was performed on a QuantStudio 6 Flex Real-Time PCR System (Applied Biosystems). Data were analysed using the Double Delta Ct method to determine fold change expression changes between samples. The number of mice per group analysed was as follows: for CD11b-positive cell fractions: female APP23 $n = 5$, male APP23 $n = 5$; for whole brain and CD11b-negative samples: female APP23 $n = 3$, male APP23 $n = 3$.

### Histology

Formalin-fixed and sucrose-treated hemispheres were frozen and cut coronally in serial sections at 40 μm using a cryostat (Thermo Scientific HM 560). Sections were kept in cryoprotectant (0.65 g NaH$_2$PO$_4$ × H$_2$O, 2.8 g Na$_2$HPO$_4$ in 250 ml ddH$_2$O, pH 7.4 with 150 ml ethylene glycol, 125 ml glycerine) at 4°C until staining. For immunohistochemistry, sections were washed in PBS, mounted on SuperFrost Plus slides (R. Langenbrink), dried and blocked in PBS with 0,3% Triton X-100 and 10% normal goat serum (NGS) for 1 h, before incubation with Aβ-specific antibody anti-4G8 targeting aa. 17–24 of human Aβ (1:1,000 dilution, Covance, SIG39320) or astrocyte-specific antibody anti-GFAP (1:1,000 dilution, Dako, Z0334) over night at 4°C in PBS with 0.3% Triton X-100 and 5% NGS. Next, sections were quenched with 0.5% H$_2$O$_2$ for 30 min at room temperature, washed and then incubated with peroxidase-conjugated goat anti-mouse secondary antibody (1:100 dilution, Dianova, 115-035-068) or peroxidase-conjugated goat anti-rabbit secondary antibody (1:100 dilution, Jackson ImmunoResearch, 111-035-003) in PBS with 0.3% Triton X-100 and 5% NGS for 1 h at room temperature. The staining was developed using diaminobenzidine (DAB) substrate (Sigma-Aldrich). Sections were counterstained with matured haematoxylin, followed by signal development in tap water. Subsequently, sections were dehydrated in ascending ethanol concentrations (70, 80, 96, 100%) and xylene and embedded with

hydrophobic mounting medium (Roti Histokitt, Carl Roth). Immunofluorescent co-labelling of anti-4G8 (1:1,000 dilution, Covance, SIG39320) and anti-BACE1 (1:500 dilution, Abcam, ab108394) was performed as above with primary antibody incubation for 48 h at 4°C and incubation with secondary antibodies Alexa Fluor 568 goat anti-rabbit IgG (H+L) (1:500 dilution, Invitrogen, A11011) and Alexa Fluor 647 goat anti-mouse IgG (H+L) (1:500 dilution, Invitrogen, A21236) for 2 h at room temperature. Immunofluorescent labelling of anti-4G8 (1:1,000 dilution, Covance, SIG39320), anti-Iba1 (1:500 dilution, Wako, 019-19741) and marker of activated microglia anti-Clec7a (Dectin-1) (1:30 dilution, InvivoGen, mabg-mdect) was modified to include a 10 min permeabilisation step at room temperature in TBS with 0.2% Triton-X. For all following steps, no Triton-X was added to the solutions. Blocking was performed in TBS with 10% NGS and antibody incubations in TBS with 5% NGS. Again, primary antibodies were incubated for 48 h at 4°C and secondary antibodies Alexa Fluor 488 goat anti-mouse IgG (H+L) (1:500 dilution, Invitrogen, A11001), Cy™3 AffiniPure Donkey Anti-Rat IgG (H+L) (1:500 dilution, Jackson ImmunoResearch, 712-165-153) and Alexa Fluor 647 goat anti-rabbit IgG (H+L) (1:500 dilution, Invitrogen, A21244) were incubated for 2 h at room temperature. Following fluorescent immunostaining, sections were counterstained with DAPI (1:2,500 dilution, Sigma-Aldrich, 10236276001) and embedded with fluorescence mounting medium (Dako, S3023).

Congo Red staining [37] was performed on mounted and dried sections counterstained with matured haematoxylin. Sections were incubated in stock solution I (0.5 M NaCl in 80% ethanol, 0.01% hydrous NaOH) for 20 min and in stock solution II (8.6 mM Congo Red in stock solution I, 0.01% NaOH) for 45 min. Subsequently, sections were rinsed in 80% EtOH and xylene and embedded with hydrophobic mounting medium.

For staining with pentameric formyl thiophene acetic acid (pFTAA) [36], sections were washed in PBS, stained for 30 min with 2 μg/ml pFTAA in PBS and counterstained with 4′,6-diamidino-2-phenylindole (DAPI) (1:5,000, Sigma-Aldrich). Sections were embedded in fluorescent mounting medium (Agilent).

### Stereological analysis of Aβ plaque burden and cortical astrocyte number

For quantifying Aβ plaque load and astrocytes number, the Stereo Investigator system (MBF Bioscience) mounted on an Olympus BX53 microscope, equipped with the QImaging camera COLOR 12 BIT and a stage controller MAC 6000 was used. Quantification of cortical area covered by 4G8-positive or Congo Red-positive Aβ plaques was undertaken using the Stereo Investigator 64-bit software (MBF Bioscience) (settings: 10× objective, counting frame 90 × 90 μm, scan grid size 450 × 450 μm, Cavalieri grid spacing 10 μm). For counting cortical astrocytes, the Optical Fraction Fractionator tool of the Stereo Investigator 64-bit software (MBF Bioscience) was used (settings: 40× objective, counting frame 75 × 75 μm, scan grid size 500 × 500 μm). The values from "Estimated Population using User Defined Section Thickness" and "Measured Volume (μm³)" were divided and used to calculate the number of cells per cortical volume. For quantification of pFTAA-positive plaques, the Olympus cellSens Dimension software was used. Sections were exposed at 400 ms, and a region of interest

(ROI) was selected around the cortex. The image was converted to grey scale, and the same threshold was applied to obtain the area fraction of pFTAA-positive signal (%). For each stain, 10 serial coronal sections per brain were used for analysis. The number of mice per group analysed was as follows: female APP23 $n = 8$, female APP23p40$^{-/-}$ $n = 10$, male APP23 $n = 10$ and male APP23p40$^{-/-}$ $n = 8$.

### Analysis of BACE1/4G8 ratio

Images of BACE1/4G8 co-labelled sections were taken on an Olympus BX53 microscope, equipped with the QImaging camera COLOR 12 BIT and controlled by the Olympus cellSens Dimension software. Per animal, images were taken from 10 serial coronal sections and three regions per section. In ImageJ, we performed a batch conversion of raw TIFF images as contrast-optimised, greyscale JPEG files, and using a custom R script, we extracted the generated histograms and used these to calculate the respective proportion of stained and unstained pixels. A fixed analysis threshold was chosen based on variance and mean image intensity of all analysed images belonging to the BACE1 antibody and 4G8 antibody staining, respectively. Quality of the histological stainings and image material was estimated by a cross-comparison of each image's characteristics to (i) all other images of the same animal and (ii) all other images of the same experimental group and all images that did not meet the defined acceptable range of 2*SD were excluded from downstream analysis (10.7% of images total). The median BACE1/4G8 ratio of 16–30 images per animal was taken for analysis, and the mean and SEM of all animals from one group was plotted. The number of mice per group analysed was as follows: female APP23 $n = 8$, female APP23p40$^{-/-}$ $n = 10$, male APP23 $n = 10$ and male APP23p40$^{-/-}$ $n = 8$. The R script has been deposited on Github (https://github.com/eedep/Image-processing).

### Quantification of Clec7a-positive plaque-associated microglia and Aβ uptake

Three-dimensional image stacks (1 μm step size, 40× objective) of 4G8/Clec7a/Iba1-stained sections were taken on a Leica TCS SP5 confocal laser scanning microscope controlled by LAS AF scan software (Leica Microsystems, Wetzlar, Germany). Per animal, 10 serial coronal sections and three regions per section were used for analysis. The number of mice per group analysed was as follows: female APP23 $n = 8$, female APP23p40$^{-/-}$ $n = 10$, male APP23 $n = 10$ and male APP23p40$^{-/-}$ $n = 8$.

The expression levels of Iba1 and Clec7a, 4G8 positive plaques, radial intensity profiles, cell numbers and distances to the nearest plaque (Figs 2C, 5C and 7C) were quantified from maximum projections of the confocal stacks. The quantification was performed in an automated manner using custom-written ImageJ macros (segmentation) [59] and python scripts (radial profiles and other statistics), which can be found on GitHub (https://github.com/ngimber/AlzheimersWorkflow). Data were pooled by calculating the median from all images per animal and plotting the mean and SEM of all animals from one group. Data that are displayed as a histogram were binned image-wise. Histograms and radial intensity profiles were normalised (divided by its own integral) and then pooled as mentioned above.

### Segmentation (ImageJ)

Nuclei were segmented from blurred DAPI channels (Gaussian blur, sigma = 720 nm) by histogram-based thresholding (Otsu binarisation) [60] followed by watershed segmentation of the Euclidean distance map of the binary image using ImageJ. Plaques were segmented from the blurred 4G8 channel (Gaussian blur, sigma = 7.2 μm) followed by Otsu binarisation. Only objects above 720 μm² were regarded as plaques.

### Quantification (Custom Python scripts)

The mean intensities within segmented nuclear regions (s. above) were used as a measure for Iba1 and Clec7a expression levels. Cells were classified as Iba1-positive/-negative by auto-thresholding (Otsu's method on all cell-specific expression levels within one image). Only Iba1-positive cells were used for further analysis (e.g. Clec7a expression levels, cell numbers, distances to the nearest plaque and radial intensity profiles). The size of the nearest plaques was determined for each cell based on the segmented regions mentioned above. Radial intensity profiles were calculated for all channels around the centre of mass of the segmented nucleus.

### Brain homogenisation

For analysis of protein levels, frozen hemispheres were subjected to a 4-step protein extraction protocol, using buffers with increasing stringency [34]. In brief, hemispheres were homogenised consecutively in Tris-buffered saline (TBS) buffer (20 mM Tris, 137 mM NaCl, pH = 7.6), Triton-X buffer (TBS buffer containing 1% Triton X-100), SDS buffer (2% SDS in ddH$_2$O) and FA (70% formic acid in ddH$_2$O). Immediately before use, cOmplete™ Mini Protease Inhibitor Cocktail Tablets (Roche, 1 tablet per 10 ml) were added to all buffers. Initial homogenisation occurred mechanically by consecutive passing the solution through a 2-ml syringe and cannulas with decreasing diameter (G23, G27 and G30). Brain extracts were incubated 30 min on ice (except SDS homogenate, which was incubated at RT) and centrifuged at 100,000 $g$ for 1 h at 4°C. The supernatant was collected, aliquoted, snap-frozen in liquid nitrogen and stored at −80°C until further use. The remaining pellet was re-suspended in subsequent buffers. Protein concentrations of each fraction were determined using the Quantipro BCA Protein Assay Kit (Pierce) according to the manufacturer protocol using the Photometer Tecan Infinite® 200M (Tecan).

### ELISA analysis

An IL-12/IL-23 total p40 enzyme-linked immunosorbent assay (ELISA) (eBioscience) was performed according to manufacturer's instructions. Undiluted TBS brain homogenate was analysed in duplicate. Absorption was read at 450 and 570 nm (for wavelength correction) on a microplate reader (Infinite® 200M, Tecan) and analysed using the Magellan Software (Tecan).

### Quantification of Aβ levels

Brain extracts of all TBS, Triton-X and SDS fractions were analysed for Aβ40 and Aβ42 levels using the 96-well MultiSpot Human 6E10 Aβ Triplex Assay Kit (Meso Scale Diagnostics, MSD). In brief, samples were analysed in duplicate and were diluted to fit the standard curve (Aβ Peptide 3-Plex). After blocking the MSD plate with

### Quantification of cytokines

Pro- and anti-inflammatory markers [IFNγ, IL-10, IL-1β, IL-2, IL-4, IL-5, IL-6, TNF-α, CXCL1 (KC/GRO)] were analysed in the TBS fraction of brain homogenates and plasma samples using the 96-well 10-plex Pro-inflammatory Panel 1 (mouse) Mesoscale Kit according to manufacturer's instructions (MSD In brief, undiluted TBS homogenate, plasma samples diluted 1:100 or the calibrator was added in duplicate to the MSD plate and incubated for 2 h. After washing in 0.05% Tween-20 in PBS, the detection antibody solution was added and incubated for further 2 h. After washing the plate with 0.05% Tween-20 in PBS, 2× Reading Buffer was added to the wells and the plate was analysed on a MS6000 machine (MSD). The number of mice per group analysed was as follows: female APP23 $n = 7$, female APP23p40$^{-/-}$ $n = 8$, male APP23 $n = 10$ and male APP23p40$^{-/-}$ $n = 8$.

### Western blot analysis

For the quantification of BACE1, neprilysin and insulin-degrading enzyme (IDE), the Triton-X fraction of brain homogenates (30 μg/lane) was separated by SDS–PAGE using 10% Tris-Glycine gels. For quantifying 6E10, the SDS fraction of brain homogenates (30 μg/lane) was separated by SDS–PAGE using Novex™ 10–20% Tricine protein gels (Invitrogen, EC66255BOX). Proteins were transferred by wet blotting onto a nitrocellulose membrane.

Membranes were blocked with 3% milk powder and stained with the anti-Aβ 6E10 antibody (1:2,000, BioLegend, 803002), anti-BACE1 (1:1,000, Abcam, ab108394), anti-Neprilysin (CD10) (1:500, Invitrogen, PA5-29354), anti-IDE (1:1,000, Merck, PC730) and either anti-β-Actin (1:50,000, Sigma, A1978) or anti-GAPDH (1:500, Merck, MAB374). Blots pre-stained with anti-IDE were stripped using the Abcam Mild Stripping protocol in order to re-stain with anti-Neprilysin. Secondary staining was performed using the ECL HRP-linked anti-mouse antibody (1:5,000, GE Healthcare, NA931) or ECL HRP-linked anti-rabbit antibody (1:5,000, GE Healthcare, NA934) and for visualisation of the bands the SuperSignal® West Femto Chemiluminescent Substrate (Thermo Fisher) for the detection of horseradish peroxidase activity was used. For quantification, the intensities of the corresponding bands for each protein were determined with ImageJ and the amount of the respective protein was normalised either to the β-actin or GAPDH protein content. For the analysis of BACE1, Neprilysin and IDE, some of the loaded Triton-X samples did not contain enough protein sufficient for analysis (based on GAPDH content). These specific samples were removed from the analysis.

### Filter retardation test

A native filter test was applied to analyse size and stability of Aβ aggregates on non-denatured samples [modified from 38]. In brief,

brain homogenates from all four protein fractions (10 μg total protein per dot) were filtered in triplicate through a 0.2-μm cellulose acetate membrane. Synthetic pre-fibrillar Aβ was used as a positive control and NSP buffer (10 mM $K_3PO_4$, 10 mM NaCl pH 7.4) as a negative control. Filters were washed in PBS and incubated with the 6E10 antibody (1:2,000, BioLegend, 803002), followed by a mouse secondary HRP-conjugated antibody (Sigma, A0168), to allow chemiluminescent detection of the aggregated proteins remaining on the filter. Membranes were exposed for 1 min, and signals were analysed using the Aida program. A detailed setup of the membrane can be found in Appendix Fig S2.

**Statistics, data analysis, study design and data availability**

*General*

Data were generated based on multiple exploratory histological and biochemical analyses aimed at generating hypotheses and biostatistical planning for future confirmatory studies. Data analysis, processing, descriptive and formal statistical testing were done according to the current customary practice of data handling using Excel 2016, GraphPad PRISM 5.0, ImageJ, Python 3.7.4 and R version 3.5.1 "Feather Spray" (code available via Github). To display data in a consistent manner, graphs were generated using PRISM, while correlation graphs were done by the use of ggplot2 in R. All data generated or analysed during this study are included in this article.

*Statistics*

Student's *t*-test was used for pairwise comparison between two experimental groups. For Clec7a column analysis, a Bonferroni correction for each single bin was applied. Pearson *r*-value and *P*-value for correlations were identified using correlation analysis. One-way ANOVA testing was applied for comparison of more than two groups, with post hoc analysis using Tukey's multiple comparison test. Statistical significance is indicated as follows: $*P \leq 0.05$, $**P \leq 0.01$ and $***P \leq 0.001$.

**Expanded View** for this article is available online.

## Acknowledgements

This work was supported by the Deutsche Forschungsgemeinschaft (DFG, German Research Foundation) under Germany's Excellence Strategy—EXC-2049—390688087, as well as under SFB TRR 43, SFB TRR 167 and HE 3130/6-1 to F.L.H., SFB 958/Z02 to J.S., by the German Center for Neurodegenerative Diseases (DZNE) Berlin, and by the European Union (PHAGO, 115976; Innovative Medicines Initiative-2; FP7-PEOPLE-2012-ITN: NeuroKine). We are indebted to Eilís Perez for generating correlation graphs and to Kiara Freitag for assistance with graphical illustrations. Synopsis image was created with Biorender.com.

## Author contributions

PE and JO performed experiments and analysed data; EEW and AB performed filter retardation tests and analyses; EB performed histological stainings and generated confocal images; GY-D performed stereological analyses; BCR wrote the custom R script for analysing the BACE1/4G8 ratio; NG and JS analysed confocal images for plaque-associated microglia and Clec7a intensity as well as 4G8-positive microglia; FLH and SP designed and supervised the study; PE prepared figures. All authors wrote, revised and approved the manuscript.

## Conflict of interest

The authors declare that they have no conflict of interest.

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
