## [Review Process File · EMBO Reports]

Interleukin-12/23 deficiency differentially affects pathology in male and female Alzheimer's disease-like mice

Pascale Eede, Juliane Obst, Eileen Benke, Genevieve Yvon-Durocher, Bernhard C. Richard, Niclas Gimber, Jan Schmoranz, Annett Böddrich, Erich E. Wanker, Stefan Prokop, Frank L. Heppner

Review timeline:

Submission date:	21 May 2019
Editorial Decision:	9 July 2019
Revision received:	15 November 2019
Editorial Decision:	11 December 2019
Revision received:	19 December 2019
Accepted:	8 January 2020

Editor: Esther Schnapp

Transaction Report:

1st Editorial Decision

9 July 2019

Thank you for the submission of your manuscript to EMBO reports. We have now received the enclosed referee reports on it.

As you will see, while the referees acknowledge that the gender-specific effects are interesting, both referees 2 and 3 point out that in the absence of a little more mechanistic insight the paper will not be a good fit for EMBO reports. Both referees suggest that microglia activation around the plaques should be analyzed, in addition to a few other suggestions.

Given these constructive comments, I would thus like to invite you to revise your manuscript with the understanding that the referee concerns must be fully addressed and their suggestions taken on board. Please address all referee concerns in a complete point-by-point response. Acceptance of the manuscript will depend on a positive outcome of a second round of review. It is EMBO reports policy to allow a single round of revision only and acceptance or rejection of the manuscript will therefore depend on the completeness of your responses included in the next, final version of the manuscript.

Revised manuscripts should be submitted within three months of a request for revision; they will otherwise be treated as new submissions. Please contact us if a 3-months time frame is not sufficient for the revisions so that we can discuss this further. You can either publish the study as a short report or as a full article. For short reports, the revised manuscript should not exceed 27,000 characters (including spaces but excluding materials & methods and references) and 5 main plus 5 expanded view figures. The results and discussion sections must further be combined, which will help to shorten the manuscript text by eliminating some redundancy that is inevitable when discussing the same experiments twice. For a normal article there are no length limitations, but it should have more than 5 main figures and the results and discussion sections must be separate. In both cases, the entire materials and methods must be included in the main manuscript file.

Regarding data quantification, please specify the number "n" for how many independent

experiments were performed, the bars and error bars (e.g. SEM, SD) and the test used to calculate p-values in the respective figure legends. This information must be provided in the figure legends. Please also include scale bars in all microscopy images.

5) a complete author checklist, which you can download from our author guidelines <https://www.embopress.org/page/journal/14693178/authorguide>. Please insert information in the checklist that is also reflected in the manuscript. The completed author checklist will also be part of the RPF.

6) Please note that all corresponding authors are required to supply an ORCID ID for their name upon submission of a revised manuscript (<https://orcid.org/>). Please find instructions on how to link your ORCID ID to your account in our manuscript tracking system in our Author guidelines <https://www.embopress.org/page/journal/14693178/authorguide#authorshipguidelines>

7) We would also encourage you to include the source data for figure panels that show essential data. Numerical data should be provided as individual .xls or .csv files (including a tab describing the data). For blots or microscopy, uncropped images should be submitted (using a zip archive if multiple images need to be supplied for one panel). Additional information on source data and instruction on how to label the files are available at <https://www.embopress.org/page/journal/14693178/authorguide#sourcedata>.

I look forward to seeing a revised version of your manuscript when it is ready. Please let me know if you have questions or comments regarding the revision.

REFEREE REPORTS

Referee #1:

Eede, Obst, et al report important gender differences in the APP23 mice, a slow model of AD. Specifically, female mice show more amyloid formation; moreover, the authors also investigate how gender differences impact the effects of IL-12p40 deficiency on amyloidosis. They show that in male mice, IL-12p40 deficiency reduces plaque burden, whereas in female mice it reduced soluble A β species.

This study extends previous work on the impact of gender in amyloidosis and shows an important gender difference in the impact of IL-12/IL-23 signaling in the response to amyloidosis. I find the paper highly significant, as it highlights the importance of gender considerations when studying AD and, particularly, the role of immune responses to AD. Moreover, as opposed to previous papers, this paper studies gender differences in a slow AD model (APP23), which may be more physiological than previously examined fast models of AD. Thus, the paper provides significant advances to the field. It is extremely carefully performed, very detailed and well written. I think it will be extremely impactful in the AD field and neurodegeneration, in general.

I have only one minor point that the authors may consider. If mouse tissue is still available, the authors may consider evaluating neurite dystrophy by staining for NT-APP or LAMP1. This analysis would directly indicate the impact of the amyloid on neuronal functions.

Referee #2:

The authors expand on previous published data showing that deletion of IL12p40 mitigates amyloid plaque deposition, demonstrating this in a different mouse model (APP23, slower plaque formation than the previous APPS1), and demonstrating an interesting effect of sex. Namely, they find that IL12p40 deletion has less impact on plaques in female APP23 mice, which have more plaque deposition and in particular more fibrillar plaques.

The paper has several strengths. The characterization of male vs. female APP23 at advanced ages is very thorough in terms of A β species. The effect of IL12p40 deletion is impressive, and this represents a viable therapeutic target, and is thus valuable. Finally, the sex difference in the effect of IL12p40 is interesting and has important implications.

However, there are some concerns:

1. The paper is not particularly novel. The effects of sex on plaques are well known, as are the effects of IL12p40 deletion. While this paper extends these findings to a new model and examines sex, these are modest advances. Addressing criticism 2 may alleviate this.
2. There is very little mechanistic insight in the paper. Though the authors invoke a possible effect on microglia, there is no analysis of the effects of IL12p40 deletion on glial activation/interaction. There is analysis of some cytokines in tissue, of which only CXCL1 is different, but the paper would be much stronger if some morphologic analysis of microglial activation around plaques with Iba1/Cd68/other markers were performed, as well as staining for astrocyte activation. It remains unclear if this is a differential effect of IL12/23 on female vs. male microglia, or if the effects observed are simply due to the different plaque makeup in females (more fibrillar A β). The authors discuss this a bit, but analysis of the degree of peri-plaque microglial activation/lysosome expression in female vs. male IL12p40 KO mice could help determine if there is a differential response to plaques. In vitro A β uptake assays in male vs. female microglia stimulated with IL12/23 would also be illustrative, but perhaps beyond the scope of a Report.

In general, this is a technically strong paper, but the novelty of the observation does not make up for the lack of mechanistic insight. Thus, in my opinion, a bit more detailed analysis of glial responses is needed.

Referee #3:

Gender-specific pathology is differentially affected by interleukin-12/23 subunit deficiency in male and female Alzheimer's disease-like mice

Pascale Eede¹, {section sign}, Juliane Obst^{1,2}, {section sign}, Annett Böddrich³, Erich E. Wanker³, Stefan Prokop^{1,4}, {section sign}, Frank L. Heppner^{1,5,6,7}, {section sign},*

This paper follows on from vom Berg et al in 2012 by examining the impact of IL-12p40 deletion in a model of Alzheimer's disease. While the first paper looked at APP/PS1 mice, this current article examines the impact in APP23 mice, a slower progressing animal model of disease.

The authors demonstrate that deletion of IL-12p40 is protective in male APP23 mice only, reducing plaque burden in this group alone.

This article is well written, and the results on gender differences are very timely however, this reviewer thinks that more experiments are needed to determine the reason behind the sex-dependent difference in their findings and more mechanistic data would improve the impact of this finding.

- 1) The mice used are 21 months old, this is a very late stage for mice, and it would be interesting to know if the mortality is the same in male and females. Are we already seeing a survivor effect?
- 2) In the vom Berg paper, it states that equal numbers of male and female mice are used, yet no gender differences were observed in the APP/PS1 mice after IL-12p40 deletion (even though it has been reported that there are gender differences in plaque load in APP/PS1 mice). It would be interesting to hear the authors take on this finding in contrast with the APP23 model and the findings in this current paper.
- 3) It would be nice to clearly demonstrate that in APP23 mice, the IL-12p40 is coming from microglia, the authors refer to finding from a different mouse model (APP/PS1 mice) on this point.
- 4) It would be great to see staining of the microglia around the amyloid plaques in the male vs females. Are the microglia taking up more Abeta? Is the deposition reduced in the ko's? What are the levels of BACE? What is the effect on Abeta degradation enzymes e.g. IDE or Nephilysin? This information would help to understand the sex differences observed in the present study
- 5) Can the authors rule out a peripheral effect of IL-12p40 deletion? Are there any changes to the peripheral inflammation? The microbiome?

1st Revision - authors' response

15 November 2019

Point-by-point-reply to referees' comments:**Referee #1:**

“Eede, Obst, et al report important gender differences in the APP23 mice, a slow model of AD. Specifically, female mice show more amyloid formation; moreover, the authors also investigate how gender differences impact the effects of IL-12p40 deficiency on amyloidosis. They show that in male mice, IL-12p40 deficiency reduces plaque burden, whereas in female mice it reduced soluble Ab species.

This study extends previous work on the impact of gender in amyloidosis and shows an important gender difference in the impact of IL-12/IL-23 signaling in the response to amyloidosis. I find the paper highly significant, as it highlights the importance of gender considerations when studying AD and, particularly, the role of immune responses to AD. Moreover, as opposed to previous papers, this paper studies gender differences in a slow AD model (APP23), which may be more physiological than previously examined fast models of AD. Thus, the paper provides significant advances to the field. It is extremely carefully

performed, very detailed and well written. I think it will be extremely impactful in the AD field and neurodegeneration, in general.

I have only one minor point that the authors may consider. If mouse tissue is still available, the authors may consider evaluating neurite dystrophy by staining for NT-APP or LAMP1. This analysis would directly indicate the impact of the amyloid on neuronal functions.”

Reply: We thank the reviewer for their positive evaluation of our manuscript and agree that evaluating direct or indirect markers of neurite dystrophy would add to the already existing data. To do so, we immunostained tissue sections with BACE1, which recently has been suggested to act as a surrogate marker of neuritic dystrophy (Zhao *et al.*, 2007, *J Neurosci*; Sadleir *et al.*, 2016, *Acta Neuropathol*; Peters *et al.*, 2019, *EMBO J*; Leyns *et al.*, 2019, *Nat Neurosci*), and co-labelled with the 4G8 antibody to stain Ab. Since differences in plaque sizes are a potential confounding element when assessing the amount of plaque-associated BACE1-immunoreactive neurites, we calculated the ratio of the area covered by BACE1 to the area covered by 4G8-positive Ab. The respective results are now included in the revised manuscript in *Fig. 2A, 5A & 7A*. In summary, these additional data reveal no differences in plaque-associated BACE1 immunoreactivity between male and female APP23 mice harbouring or lacking IL12p40.

Reviewer #2:

“The authors expand on previous published data showing that deletion of IL12p40 mitigates amyloid plaque deposition, demonstrating this in a different mouse model (APP23, slower plaque formation than the previous APPPS1), and demonstrating an interesting effect of sex. Namely, they find that IL12p40 deletion has less impact on plaques in female APP23 mice, which have more plaque deposition and in particular more fibrillar plaques.

The paper has several strengths. The characterization of male vs. female APP23 at advanced ages is very thorough in terms of Abeta species. The effect of IL12p40 deletion is impressive, and this represents a viable therapeutic target, and is thus valuable. Finally, the sex difference in the effect of IL12p40 is interesting and has important implications.

However, there are some concerns:

1. The paper is not particularly novel. The effects of sex on plaques are well known, as are the effects of IL12p40 deletion. While this paper extends these findings to a new model and examines sex, these are modest advances. Addressing criticism 2 may alleviate this.

2. There is very little mechanistic insight in the paper. Though the authors invoke a possible effect on microglia, there is no analysis of the effects of IL12p40 deletion on glial activation/interaction. There is analysis of some cytokines in tissue, of which only CXCL1 is different, but the paper would be much stronger if some morphologic analysis of microglial activation around plaques with Iba1/Cd68/other markers were performed, as well as staining for astrocyte activation. It remains unclear if this is a differential effect of IL12/23 on female vs. male microglia, or if the effects observed are simply due to the different plaque makeup in females (more fibrillar Abeta). The authors discuss this a bit, but analysis of the degree of peri-plaque microglial activation/lysosome expression in female vs. male IL12p40 KO mice could help determine if there is a differential response to plaques. In vitro Abeta uptake assays in male vs. female microglia stimulated with IL12/23 would also be illustrative, but perhaps beyond the scope of a Report.

In general, this is a technically strong paper, but the novelty of the observation does not make up for the lack of mechanistic insight. Thus, in my opinion, a bit more detailed analysis of glial responses is needed.”

Reply: We agree with the reviewer's observation that analysing both astrocyte and microglia characteristics could help in understanding our findings relating to plaque pathology in the herein used AD-like mouse models. When quantifying GFAP-positive cortical astrocytes (cells/mm³) we saw increased astrocyte numbers in female APP23 mice compared to male mice, which directly correlated to the increased 4G8- and Congo Red-positive plaque load in female mice (Fig. 2B; Fig. EV1A in revised manuscript). Upon IL12p40 deletion, we could not detect any differences in astrocyte number in either gender (Fig. 5B & 7B in revised manuscript).

In order to assess microglial characteristics, we quantified plaque-associated microglia using the microglia marker Iba1 and their expression of the activation marker Clec7a (Fig. 2C, 5C & 7C in revised manuscript). This analysis revealed no gender-specific differences or an effect of IL12p40 deletion upon the number of plaque-associated microglia or the presence of Clec7a-positive activated microglia. Instead of studying microglial A β uptake *in vitro*, as suggested by this referee, we performed radial intensity profiling on confocal images of brain tissue to measure 4G8-positive signal within Iba1-positive microglia as an indicator of microglial A β uptake. This analysis allowed depicting 4G8-positive amyloid intensity peaks inside the cell (~4 μ m) illustrating A β uptake by microglia, yet there were no differences in intracellular A β levels between male and female APP23 mice harbouring or lacking IL12p40 (Fig. 2D, 5D & 7C in revised manuscript).

Reviewer #3:

“This paper follows on from vom Berg et al in 2012 by examining the impact of IL-12p40 deletion in a model of Alzheimer's disease. While the first paper looked at APP/PS1 mice, this current article examines the impact in APP23 mice, a slower progressing animal model of disease.

The authors demonstrate that deletion of IL-12p40 is protective in male APP23 mice only, reducing plaque burden in this group alone.

This article is well written, and the results on gender differences are very timely however, this reviewer thinks that more experiments are needed to determine the reason behind the sex-dependent difference in their findings and more mechanistic data would improve the impact of this finding.

- 1) *“The mice used are 21 months old, this is a very late stage for mice, and it would be interesting to know if the mortality is the same in male and females. Are we already seeing a survivor effect?”*

Reply: In order to answer the reviewer's question, we have plotted the age of death (natural or by use in experiment) with respect to gender (Fig. R1). It becomes apparent from this qualitative analysis that male APP23 mice do not tend to die earlier than female mice and are similarly available for the use in experiments requiring aged animals. In our study looking at APP23 and APP23p40^{-/-} mice, we also did not have a drop-out of mice from the experimental groups assigned at early ages of the mice.

Figure R1. Qualitative representation of age of death (in days) from mice of the APP23 strain in our lab between 2016-2019. The left scatter depicts animals, which died from natural cause whilst the right scatter represents animals that were sacrificed and/or used in an experiment. Colour-coded symbols representing male and female mice show no substantial gender differences with respect to the death rate across all time points. Fig. R1 is only part of this rebuttal letter.

- 2) “In the vom Berg paper, it states that equal numbers of male and female mice are used, yet no gender differences were observed in the APP/PS1 mice after IL-12p40 deletion (even though it has been reported that there are gender differences in plaque load in APP/PS1 mice). It would be interesting to hear the authors take on this finding in contrast with the APP23 model and the findings in this current paper.”

Reply: We agree with the reviewer that gender differences not only appear in the APP23 mouse strain but also in APPPS1 mouse strains. The APPPS1 mouse model used in the study by vom Berg *et al.*, (2012, Nat Med) was the APPPS1-21 strain first described by Radde *et al.* (2006, EMBO Rep). Here, analyses of gender-specific A β plaque pathology are contradictory. A study by Ulrich *et al.* (2014, Mol Neurodegener) showed increased cortical plaque burden and PBS-soluble A β_{1-40} and A β_{1-42} in female APPPS1-21 mice at 3 months compared to age-matched male mice. On the other hand, Dodiya *et al.* (2019, J Exp Med) described higher cortical plaque burden in male mice compared to female mice at 7 weeks of age, yet these gender differences were not seen any more at 3 months. In Fig. R2, we depict the data on cortical 4G8-positive plaque burden in 4 month old APPPS1 and APPPS1p40^{-/-} published by vom Berg *et al.* (2012, Nat Med) with a focus on the gender of each mouse.

This confirms that our previously published data on the impact of IL12p40 deletion in APPPS1 mice were not influenced by differences between genders and supports the notion that no differences in cortical A β plaque load exist at 4 months in APPPS1-21 mice.

Figure R2. Gender-specific analysis of 4G8-positive cortical plaque load in APPPS1 and APPPS1xp40^{-/-} mice. Data points taken from the publication by vom Berg *et al.* (2012, *Nat Med*). Fig. R2 is only part of this rebuttal letter.

It is important to note, however, that several currently used APPPS1 mouse strains differ (i) in the Presenilin mutation used for generating the respective transgenic mouse strain, as well as (ii) in the promoter used for overexpression of transgenes. APPPS1-21 mice used in our previous studies harbour the Thy1 promoter, while transgene expression in other, commonly used APPPS1 models is driven by the *Prnp* promoter, which may not only account for differences in the quantity and region of transgene expression, but may also have distinct gender-specific effects. Along this line, it is interesting to learn that several previous publications demonstrate substantial gender differences in APPPS1 mouse strains using alternative PS1 mutations and promoter constructs. For example, APPPS1 mice harbouring the PSEN1 A246E mutation (Borchelt *et al.*, 1997, *Neuron*) also show increased A β pathology in female mice at 4, 12 and 17 months of age (Wang *et al.*, 2003, *Neurobiol Dis*). Another very commonly used model is the APPPS1dE9 line (Jankowsky *et al.*, 2004, *Biomol Eng*), where increased pathology has been observed in female mice between 9-12 months of age (Gallagher *et al.*, 2013, *Neurodegener Dis*; Li *et al.*, 2015, *Lab Anim*; Jiao *et al.*, 2016, *Neurotox Res*).

Taken together, we assume that the gender-specific effect in APP23 mice described by us – at least at the investigated age of 21 months – is likely a result of differences in the composition of A β amyloid including A β plaques as well as in disease progression between males and females.

- 3) “It would be nice to clearly demonstrate that in APP23 mice, the IL-12p40 is coming from microglia, the authors refer to finding from a different mouse model (APP/PS1 mice) on this point.”

Reply: This is a most valid aspect and we have now added a more thorough analysis of IL12p40 expression (*Il12b* gene) in APP23 mice (*Fig. 3A* in revised manuscript). Specifically, we isolated RNA (i) from whole brain, (ii) from microglia sorted using CD11b-positive microbeads as well as (iii) the CD11b-negative cell fraction from both male and female APP23 mice. qPCR analysis for the *Il12b* gene revealed that only the microglia-positive cell fraction showed *Il12b* expression. No *Il12b* expression was detected in whole brain and CD11b-negative cell fractions. Microglial *Il12b* expression also did not differ between male and female APP23 mice, which also correlates with IL12p40 protein expression as measured by ELISA (see *Fig. 3B* in revised manuscript).

- 4) *“It would be great to see staining of the microglia around the amyloid plaques in the male vs females. Are the microglia taking up more Abeta? Is the deposition reduced in the ko's? What are the levels of BACE? What is the effect on Abeta degradation enzymes e.g. IDE or Neprilysin? This information would help to understand the sex differences observed in the present study.”*

Reply: With respect to visualising microglia around A β plaques in male and female APP23 mice, we have now added a quantification of plaque-associated microglia using the myeloid cell marker Iba1 as well as an analysis of the microglial activation marker Clec7a (now shown in *Fig. 2C, 5C & 7C* in revised manuscript). This analysis revealed no gender-specific differences or an effect of IL12p40 deletion upon the number of activated plaque-associated microglia.

We also agree with this referee that measuring microglial A β uptake is also of interest and refer to our reply to referee #2 (see above): in brief, radial intensity profiling on confocal images of brain tissue measuring 4G8-positive signals within Iba1-positive microglia as an indicator of microglial A β uptake did not reveal differences in intracellular A β levels between male and female APP23 mice harbouring or lacking IL12p40 (*Fig. 2D, 5D & 7C* in revised manuscript).

We also performed Western Blot analysis of BACE1, IDE and Neprilysin comparing the effect of IL12p40 deletion in male and female APP23 mice. The results can be found in *Fig. EV2B & EV3B* of the revised manuscript. Expression levels of all three proteins were unchanged in APP23 versus APP23p40^{-/-} animals indicating that the sex-specific effect of IL12p40 deletion upon A β pathology appears not to be regulated by differences in A β processing or degradation.

- 5) *“Can the authors rule out a peripheral effect of IL-12p40 deletion? Are there any changes to the peripheral inflammation? The microbiome?”*

This referee comes up with an interesting point since the IL-12/IL-23 pathway also plays an important role in T-cell mediated immune responses in the periphery. By generating bone-marrow chimeric APPPS1 mice, vom Berg *et al.* (2012, Nat Med) could confirm that the IL12p40-dependent effects on plaque load were mediated by the radioresistant microglial compartment. Adding this approach to our experimental setup would have been very interesting, yet beyond the scope of this study. Another alternative to exclude peripheral effects of IL12p40 deletion would be to generate mice harbouring the deletion only in microglial cells, e.g. by crossbreeding the microglia-specific Sall1CreERT line (Inoue *et al.*, 2010, Genesis) to an IL12p40^{fllox/fllox} line – a long-lasting and involved experiment, which also is not conceivable within the realms of this study.

Given that an analysis of the microbiome – despite being fascinating – was not feasible due to the lack of collecting the respective faecal samples during the study, we rather investigated potential IL12p40-mediated effects on peripheral inflammation by analysing plasma samples

collected from mice of our study using the 10-plex Pro-inflammatory Panel 1 (mouse) Mesoscale Kit (Fig. EV2A-I, EV2C-K & EV3C-K in revised manuscript). Adding to the differences seen in brain cytokine levels between male and female APP23 mice harbouring or lacking IL12p40, we found that female APP23 mice show significantly increased IL-10 plasma levels compared to male mice. Upon IL12p40 deficiency, peripheral IFN γ levels were found to be decreased in male mice, whilst female APP23p40^{-/-} mice showed both an upregulation of IL1 β and CXCL1 as well as a downregulation of IL-5 and IL-6 in their plasma. Such differences in the cytokine milieu in brain and plasma of male and female APP23 may account for the observed gender-specific variation in AD pathogenesis. Interestingly, we also found a correlation between CXCL1 levels in the brain and both soluble and insoluble A β ₁₋₄₀ (Fig. EV1B in revised manuscript). In light of the higher numbers of cortical astrocytes in female APP23 mice, it is interesting to note that astrocyte-specific induction of CXCL1 in a mouse model of multiple sclerosis augmented disease progression via recruitment of neutrophils (Grist *et al.*, 2018, Eur J Immunol), while in an AD-like mouse model blocking the entry of neutrophils into the brain had a beneficial effect upon pathogenesis (Zenaro *et al.*, 2015, Nat Med). These novel data are now included in the revised version of the manuscript (Results and Discussion section).

2nd Editorial Decision

11 December 2019

Thank you for the submission of your revised manuscript. We have now received the comments from both referees, and both support its publication now. We can therefore in principle accept your study.

A few more minor changes are still required:

- Please send us up to 5 keywords
- Please correct the reference format. Up to 10 authors need to be listed before "et al"
- Please add page numbers to the table of content of the Appendix file
- I attach to this email a manuscript word file with comments by our data editors. Please address all comments in the final manuscript file.

EMBO press papers are accompanied online by A) a short (1-2 sentences) summary of the findings and their significance, B) 2-3 bullet points highlighting key results and C) a synopsis image that is 550x200-400 pixels large (the height is variable). You can either show a model or key data in the synopsis image. Please note that text needs to be readable at the final size. Please send us this information along with the revised manuscript.

I would like to suggest some minor changes to the abstract that needs to be written in present tense:

Pathological aggregation of amyloid- β (A β) is a main hallmark of Alzheimer's disease (AD). Recent genetic association studies have linked innate immune system actions to AD development and current evidence suggests profound gender differences in AD pathogenesis. Here, we characterize gender-specific pathologies in the APP23 AD-like mouse model and find that female mice show stronger amyloidosis and astrogliosis compared with male mice. We tested the gender-specific effect of lack of IL12p40, the shared subunit of interleukin (IL)-12 and IL-23 that we previously reported to ameliorate pathology in APPPS1 mice. IL12p40 deficiency gender-specifically reduces A β plaque burden in male APP23 mice, while in female mice a significant reduction of soluble A β ₁₋₄₀ without changes in A β plaque burden is seen. Similarly, plasma and brain cytokine levels are altered differently in female versus male APP23 mice lacking IL12p40, while glial properties are unchanged. These data corroborate the therapeutic potential of targeting IL-12/IL-23 signalling in

AD, but also highlight the importance of gender considerations when studying the role of the immune system and AD.

Please let me know if you agree with these changes.

REFEREE REPORTS

Referee #2:

The authors have made considerable efforts to address my concerns, as well as those of the other reviewers. They have now added several new analyses, including plaque-related astro- and microglial markers and BACE1 for dystrophic neurites. While none of these analyses showed a clear mechanism explaining their findings, their inquiry is much more thorough now and suggests some subtle effects for many months may be at work. Thus, despite the lack of a clear mechanism, the paper is well done and adds considerably to the field.

Referee #3:

The authors responded adequately to the critique and revised the manuscript accordingly. I think the paper is now acceptable.

2nd Revision - authors' response

19 December 2019

The authors performed all minor editorial changes.

Corresponding Author Name: Frank L. Heppner

Manuscript Number: EMBOR-2019-48530V1